# Determinants of breastfeeding attitudes of mothers in Jordan: A cross-sectional study

**Sireen M. Alkhaldi**[1]*, **Oqba Al-Kuran**[2], **Mai M. AlAdwan**[3], **Tala A. Dabbah**[3], **Heyam F. Dalky**[4], **Eiman Badran**[5]

1 Department of Family and Community Medicine, School of Medicine, The University of Jordan, Amman, Jordan, 2 Department of Obstetrics and Gynecology, School of Medicine, The University of Jordan, Amman, Jordan, 3 School of Medicine, The University of Jordan, Amman, Jordan, 4 Department of Community and Mental Health Nursing, School of Nursing, Jordan University of Science and Technology, Irbid, Jordan, 5 Department of Pediatrics, School of Medicine, The University of Jordan, Amman, Jordan

* s.alkhaldi@ju.edu.jo

**Data Availability Statement:** All relevant data are within the manuscript and its Supporting Information files.

## Abstract

Breastfeeding provides the optimal nutrition for an infant. However, breastfeeding practice is on decline globally. Attitude toward breastfeeding may determine the practice. This study aimed to examine postnatal mothers' attitude to breastfeeding and its determinants. A cross-sectional study was conducted, and data on attitude were collected using the Iowa Infant Feeding Attitude Scale (IIFAS). A convenience sample of 301 postnatal women were recruited from a major referral hospital in Jordan. Data on sociodemographic characteristics, pregnancy and delivery outcomes were collected. SPSS was used to analyze the data and identify the determinants of attitudes to breastfeeding. The mean total attitude score for participants was 65.0 ±7.15, which is close to the upper limit of the neutral attitude range. Factors associated with attitude that is positive to breastfeeding were high income (p = 0.048), pregnancy complications (p = 0.049), delivery complications (p = 0.008), prematurity (p = 0.042), intention to breastfeed (p = 0.002) and willingness to breastfeed (p = 0.005). With binary logistic regression modelling, determinants of attitude positive to breastfeeding were highest income level and willingness to breastfeed exclusively (OR = 14.77, 95%CI = 2.25–99.64 and OR = 3.41, 95%CI = 1.35–8.63 respectively). We conclude that mothers in Jordan have neutral attitude to breastfeeding. Breastfeeding promotion programs and initiatives should target low-income mothers and the general population. Policymakers and health care professionals can use the results of this study to encourage breastfeeding and improve breastfeeding rate in Jordan.

## Introduction

Breastfeeding enhances bonding between a mother and her infant after birth. It is the most valuable gift a mother can offer to her newborn. The advantages of breastfeeding for both newborns and their mothers are well documented [1]. Increasing the rates of breastfeeding alone can have the potential to save the lives of around a million children under age 5 worldwide [2,

**Funding:** The authors received no specific funding for this work.

**Competing interests:** The authors have declared that no competing interests exist.

3]. Yet, about 44% of infants 0–6 months old globally were exclusively breastfed over the period of 2015–2020 [4].

Skin to skin contact and breastfeeding immediately after birth have been reported to minimize infant mortality and promote maternal health [1]. In addition, exclusive breast feeding for six months, continued for two years and beyond with the provision of safe and appropriate complementary foods are the most important strategies for promoting child survival and health [5]. However, the benefits of breastfeeding are underestimated, and the proportion of mothers who breastfeed their newborns is way below the recommended levels globally and Jordan is no exception [4, 6]. The rates of breastfeeding (any breastfeeding) reported by the UNICEF in 2022 for infants 0 to 5 months were 25% in Jordan, compared to 40% in Egypt, 35% in Morocco, 39% in Palestine, 29% in Syria, 14% in Tunisia, 29% in Algeria, 26% in Iraq, and 53% in Iran [7].

Numerous factors influence both initiation and continuity of breastfeeding. These factors include mothers' age, income, education, mode of delivery and availability of breastfeeding counseling before, during and after delivery, and past breastfeeding experience [8–14]. In addition, health professional and birth attendant's attitudes to breastfeeding have strong impact on women's breastfeeding practice, through their willingness and ability to help and encourage the mother to initiate breastfeeding [15]. In addition, type of health care facility where delivery occurred and admission of the infant in neonatal intensive care unit (NICU) influence the process of breastfeeding by women [10, 15–18].

The decision about infant feeding seems to be made before childbirth [19].The literature have shown that multiple factors at different levels determine breastfeeding practice including demographic and socioeconomic factors, cultural norms, and perceptions to breastfeeding [20, 21]. Studies have demonstrated that mothers' attitude toward breastfeeding is a significant predictor of their intention, initiation, and continuation of breastfeeding [14, 19, 22–26]. Attitude is a mental position with regard to a fact or state, and the feeling or emotion toward the fact or state. Attitude is traditionally structured along three dimensions: cognitive, affective, and behavioral [27]. The breastfeeding behavior is directly determined by the intention to breastfeed. Intention is formed through a combination of attitudes, motivation, subjective norm, and perceived behavioral control [22]. Intention is a prior conscious decision to perform a behavior. Intention can reflect maternal commitment to infant's health [28]. Researchers in Jordan demonstrated that a mother's positive attitude toward breastfeeding is a predictor of exclusive breastfeeding [24, 29]. The study population in Khasawneh et. al. [24] was a randomly selected 344 women from the North of Jordan villages. Using questions adapted from the IIFAS. While Khasawneh et. al. [29] interviewed healthy women attending antenatal clinics in three hospitals in the North of Jordan too, using the IIFAS.

Nevertheless, women's attitude toward breastfeeding worldwide is less than optimal [30]. Studies in Spain and in Canada found positive attitude to breastfeeding among women, while more neutral attitude was noted in Poland and in Fiji [9, 31–33]. In the region, Naja et al. showed that only 9.5% of women to have positive attitude to breastfeeding in Qatar and in Lebanon. On the other hand, in Saudi Arabia, Mohammed et al found more positive attitude toward breastfeeding [34], while Abulreesh et al. reported neutral attitude to breastfeeding [35]. In Ethiopia, neutral attitude to breastfeeding was reported [36]. In Jordan, several researchers found neutral attitude to breastfeeding (neither positive to breastfeeding nor to bottle feeding) [23, 37, 38], while Khasawneh et.al. (2020) reported 72% of women to have positive attitude to breastfeeding [29].

A large body of literature indicated various factors to positively influence women's attitude to breastfeeding, including education, family income, employment, previous experience with breastfeeding and intention to breastfeed exclusively [9, 11, 19, 31, 33–35, 39]. On the contrary,

maternal employment, mother's illness, lack of father's support, cesarean section delivery, pre-term delivery, infant health status and neonatal intensive care unit NICU admission were reported to adversely impact mothers' attitude to breastfeeding [19, 23, 33–35]. Advancing better understanding of the country-specific determinants and predictors of women's attitudes toward breastfeeding is critical for development of effective health care programs, and for informing health care providers and policy makers in Jordan. However, there is scarcity in research addressing factors that determine mothers' attitude toward breastfeeding in Jordan. Hence, in this study, we sought to assess factors that determine women's attitude towards breastfeeding in Jordan.

## Methods

### Study design and setting

This observational cross-sectional, hospital-based study was conducted to examine the attitudes of mothers in Jordan toward breastfeeding. Postnatal mothers, attending the maternity ward after delivery, were recruited from a referral tertiary teaching hospital in Amman, the Capital city of Jordan, during the study period February 2021 to January 2022. This hospital is an accredited Baby-friendly hospital since June 2018 and has approximately 3000 births annually.

Despite the hospital's high patient population, data collection restricted and was time-consuming because of the Covid-19 epidemic. During the pandemic, hospital admission and patient access were regularly limited when infected patients or personnel were reported.

### Population and sample

A convenience sample was selected from the target population for the current study, which was all women attending the maternity ward after delivery, when the research team was available in the hospital, during the study period. Women who were still under the effect of anesthesia, or who seemed very tired at the time of data collection were excluded. The sample size was calculated using a Raosoft, an online sample size calculator [40]. Confidence Interval was set at 95% and margin of error at 5%. A sample size of at least 377 was needed. All women in the maternity ward were eligible to participate in the study.

### Study instrument

The questionnaire for this study consisted of three parts. The first part is the Iowa Infant Feeding Attitude Scale (IIFAS) [28], a pre-validated tool used to measure attitudes of mothers toward infant feeding. An infant is defined as a child who is less than one year of age. It includes 17 items with a 5-point Likert scale ranging from 1 (strongly disagree) to 5 (strongly agree). Eight statements indicate positive attitude toward breastfeeding and nine statements indicate positive attitude toward formula feeding. The statements indicating positive attitude to breastfeeding were reverse-scored before calculating the total scores of all items. An example for attitude is: "Formula-feeding is more convenient than breastfeeding", and "Breastfeeding increases mother-infant bonding". The total IIFAS score can range from 17 to 85 with higher scores reflecting more positive attitude towards breastfeeding. The total IIFAS is further classified as positive to breastfeeding (a score of 70–85), neutral (a score of 49–69) and positive to formula feeding (a score of 17–48). The researcher used a validated Arabic version of the IIFAS after permission from the author [41]. The tool was pilot tested on 30 participants to check for clarity, response time, any potential problems with the questionnaire in addition to

reliability. Cronbach's alpha analysis revealed good reliability of the IIFAS, with coefficient equals to 0.72. No major changes were made based on the pilot test results.

Additionally, sociodemographic characteristics of mothers, and pregnancy and delivery information which includes mode of delivery, gestational age, the presence of complications during pregnancy or delivery, admission of the baby to neonatal intensive care unit (NICU), and receiving antenatal counseling for breastfeeding were all collected. The last part consisted of five questions to assess breastfeeding practices, which was created by the researcher based on the Centers for Disease Control and Prevention (CDC) guidelines [42]. These include items about intention to breastfeed, willingness to breastfeed exclusively (feeding the baby only breastmilk), receiving counselling for breastfeeding and time of initiation of breastfeeding. All data collected were quantitative.

## Data collection procedure

The research team were trained in interviewing procedures. They collected data from postpartum mothers who attended the hospital's maternity ward after delivery. Data collection took place in the period between February 2021 and January 2022, using a structured interviewer-administered questionnaire.

The research team attended the maternity ward whenever they were allowed to do so. They approached all mothers available in the ward at the time of their visit. They introduced themselves, and offered women the opportunity to participate in the research study. Women who were willing to participate were given further details and a verbal consent was obtained. Medical students were responsible for data collection, where they asked mothers questions and mothers responded. In compliance with infection control procedures, and for avoiding paper-based questionnaires, medical students entered the answers electronically immediately, using their mobile phones or tablets. It took 10 minutes to complete each questionnaire. The response rate was 95% with 301 mothers completing the questionnaires to collect sociodemographic information and the IIFAS questions. Data related to the pregnancy and delivery outcomes were collected from the medical record. Each mother's electronic medical record was searched for information pertaining to the pregnancy and birth outcomes. Every questionnaire was checked for completeness before the research team left the ward. During data collection, appropriate infection prevention control measures and principles relevant to COVID-19 were also addressed.

Data collection required a very long time, because of the Covid-19 pandemic-related restrictions for access of the research team, recurrent closures, and because of the limited number of women admitted for delivery compared to the usual number before Covid-19 pandemic. Data collection continued for 12 months, ending up with a sample of 301 participants by January 2022.

## Ethical considerations

The ethical committee (IRB) in the University of Jordan Hospital approved this study (no. 2019–229). All study protocols complied with the guidelines established by the institution's committee on research ethics. Participation in the study was voluntary, and verbal consent was obtained from participating mothers. Mothers were assured that the service they receive would not be affected by their decision to participate in the study. Data obtained were kept confidential, and were used strictly for the purpose of this research. Confidentiality of the data was preserved, data were stored in password-protected files in the computers of the research team, and were used only for the purpose of this research.

## Analysis

The collected data were examined for completeness and consistency, and then data was cleaned. Statistical analysis were performed using Statistical Package for Social Sciences (SPSS) version 21 [43]. Descriptive statistics were used to summarize the sociodemographic personal characteristics, pregnancy and delivery variables, and infant breastfeeding-related variables. In addition, the mean and standard deviation for individual items of IIFAS, and total score for the attitude toward breastfeeding for each respondent were calculated. Independent t-test and one-way ANOVA tests were performed to assess if the total attitude score towards breastfeeding different significantly across each of the measured patient characteristics Additionally, binary logistic regression was used to detect predictors of attitude of mothers toward infant feeding. The results were expressed as adjusted odds ratio (AOR) with 95% confidence intervals (CI) and p-values. P-values of <0.05 were considered statistically significant.

## Results

Data collected from 301 mothers after delivery were included in the analysis. Table 1 presents the sociodemographic characteristics of participants in addition to pregnancy and delivery information. More than 40% of the study population were 30 years of age or older and more than 70% had up to three children (including the recently delivered infant). Over 75% of participants had a university degree or higher. A large proportion of the sample were employed (43.2%). More than 60% of mothers had a monthly family income of 450 Jordan Dinar or less, and the vast majority (93%) had health insurance. The most common pregnancy complication among mothers was gestational diabetes (16%), and 26% of mothers had preterm deliveries. More than half of all deliveries were by a cesarean section (56.2%) and 18% of infants were admitted to NICU (Table 1).

Findings regarding breastfeeding-related variables are presented in Table 2. Almost all (95.7%) of mothers intended to breastfeed their infants, but only 35.2% of them were willing to breastfeed exclusively. Only 40% of mothers received any breastfeeding counselling before or after delivery. Nevertheless, only 43% of mothers initiated breastfeeding during the first day after birth, while 28.6% did not initiate breastfeeding before discharge (Table 2).

The attitude of participants to infant feeding, as measured using the IIFAS after delivery, are demonstrated in Table 3. The statements with the highest mean were "Breastfeeding increases mother-infant bonding" and "Breast milk is more easily digested than formula". The statements with the lowest means were "Mothers who occasionally drinks alcohol should not breastfeed" and "Women should not breastfeed in public places". Overall mean attitude toward infant feeding was 4.0 ± 1.0, measured using a 5-point likert scale, and the overall total (summed) attitude score for participants was 65±7.2 (out of 85). This an attitude score that is close to the upper limit of the neutral range (49–69), indicating tendency toward a positive attitude to breastfeeding. Only 24.3% of mothers in this study demonstrating positive attitude to breastfeeding after childbirth (total attitude score ≥70 out of 85) (Table 3).

Table 4 displays the relationship between mother's total attitude toward breastfeeding and personal characteristics, pregnancy and delivery variables, using t-test and ANOVA. Results show that total attitude score toward breastfeeding was significantly higher among women with higher income (p = 0.05), who had pregnancy complications (p = 0.05), with delivery complications (p = 0.01), who had preterm infants (p = 0.04), who intended to breastfeed (p = 0.00) and who were willing to breastfeed exclusively (p = 0.01).

Table 5 presents result of logistic regression carried out to assess the factors associated with positive attitude toward breastfeeding. The overall model was statistically significant when compared to the null model, ($\chi 2$ (9) = 21.7, p = 0.01). This model explained 28.4% (Nagelkerke

**Table 1. Sociodemographic characteristics, pregnancy and delivery variables of mothers participating in this study of attitudes toward breastfeeding (N = 301).**

| Variable | Categories | N | % |
|---|---|---|---|
| **Age** | <24 | 25 | 8 |
| | 24–30 | 105 | 35 |
| | 31–35 | 103 | 34 |
| | >35 | 68 | 23 |
| **Parity** | 1 | 84 | 27.9 |
| | 2–3 | 131 | 43.5 |
| | > = 4 | 86 | 28.6 |
| **Level of Education** | Primary or lower | 10 | 3.3 |
| | Secondary level | 61 | 20.3 |
| | University degree or above | 137 | 45.5 |
| | Graduate degree | 93 | 30.9 |
| **Smoking (n = 291)** | Non-smoker | 238 | 82 |
| | smoker | 15 | 5 |
| | Previous smoker | 38 | 13 |
| **Working status** | Employed | 130 | 43.2 |
| | Unemployed | 171 | 56.8 |
| **Monthly family income [a] (n = 179)** | 0–300 JD | 32 | 17.9 |
| | 301–450 JD | 79 | 44.1 |
| | 451–650 JD | 53 | 29.6 |
| | >651 JD | 15 | 8.4 |
| **Health Insurance** | Yes | 279 | 93 |
| | No | 21 | 7 |
| **Pregnancy Complications** | No | 205 | 68.1 |
| | Gestational diabetes | 44 | 16.4 |
| | Hypertension | 17 | 5.7 |
| | Preeclampsia | 11 | 3.7 |
| | Other complications | 24 | 8 |
| **Delivery Complications** | No | 271 | 90 |
| | Yes | 30 | 10 |
| **Gestational Age (mean = 41.4 week)** | Preterm | 78 | 26 |
| | Term | 222 | 74 |
| **Mode of Delivery** | Vaginal birth | 132 | 43.9 |
| | Cesarean Section Delivery | 169 | 56.1 |
| **Admission to NICU** | No admission to NICU | 247 | 82.0 |
| | Admission to NICU | 54 | 18.0 |

[a] 1 JD equals US$ 1.4

R square) of the variance in the attitude to breastfeeding and correctly classified 82% of cases. Logistic regression results indicated that, holding other variables constant, mothers with second income level (301–450 JD's) and mothers with highest family income (>650 JD's) had 4.88 and 14.77 times the odds of having positive attitude toward breastfeeding compared to women with lowest income level (95%CI = 1.12–23.8 and, 95%CI = 2.25–99.64 respectively). Furthermore, holding other variables constant, the odds of having positive attitude toward breastfeeding was 3.4times higher among mothers who were willing to breastfeed exclusively, compared to women who were not willing to breastfeed exclusively (p = 0.01, 95%CI = 1.35–8.63).

**Table 2. Breastfeeding- related variables of mothers participating in this study of attitudes toward breastfeeding (N = 301).**

| Variable | Categories | N | % |
|---|---|---|---|
| **Intention to breastfeed** | Yes | 288 | 95.7 |
| | No | 10 | 3.3 |
| | Undetermined | 3 | 1.0 |
| **Willingness to breastfeed exclusively** | Yes | 106 | 35.2 |
| | No | 146 | 48.5 |
| | Undetermined | 49 | 16.3 |
| **Breastfeeding counseling received** | Yes/ prenatal | 60 | 20 |
| | Yes/ postnatal | 60 | 20 |
| | No | 181 | 60 |
| **Time of Initiation of breastfeeding** | Day 1 | 130 | 43.2 |
| | Day 2 | 53 | 17.6 |
| | Day 3 | 32 | 10.6 |
| | Did not initiate | 86 | 28.6 |

## Discussion

The rates of breastfeeding are on decline globally including Jordan. Mothers' attitude toward breastfeeding is a vital factor that influences breastfeeding practice. This study aimed to assess mothers' attitude toward breastfeeding in Jordan, and to identify the determinants and predictors of this attitude. Results of this study exhibit that a small proportion (24.3%) of mothers in Jordan have positive attitude toward breastfeeding. This may be due to the belief that formula

**Table 3. Item scores of mothers' postnatal attitudes to breastfeeding using the IIFAS (N = 301).**

| Attitude statement | Likert Scale Responses* (%) | | | | | Item Mean ±SD |
|---|---|---|---|---|---|---|
| | 1 | 2 | 3 | 4 | 5 | |
| Benefit of breast milk last only until weaning ** | 48.8 | 4.0 | 11.6 | 7.3 | 27.2 | 3.4±1.7 |
| Formula milk is more convenient than breastfeeding ** | 39.8 | 4.3 | 8.3 | 4.7 | 29.2 | 3.5±1.8 |
| Breastfeeding increases mother-infant bonding | 1.0 | 0.0 | 1.3 | 0.7 | 96.0 | 4.9±0.6 |
| Breast milk is lacking in iron ** | 49.2 | 6.6 | 22.6 | 5.6 | 15.9 | 3.7±1.5 |
| Formula-fed babies are more likely to be overfed | 6.0 | 3.0 | 10.3 | 4.3 | 73.0 | 4.3±1.3 |
| Formula-feeding is the better choice if a mother plans to work outside to work ** | 12.3 | 3.0 | 19.3 | 12.3 | 43.5 | 2.3±1.4 |
| Mothers who formula feed miss one of the greatest joys of motherhood | 5.6 | 1.0 | 3.7 | 7.3 | 79.0 | 4.5±1.1 |
| Women should not breastfeed in public places ** | 10.9 | 5.0 | 12.3 | 4.3 | 63.0 | 2.0±1.5 |
| Babies fed breast milk are healthier | 4.3 | 1.3 | 3.7 | 5.7 | 84.0 | 4.6±1.0 |
| Breastfed babies are more likely to be overfed ** | 76.7 | 6.6 | 8.6 | 2.7 | 5.3 | 4.5±1.1 |
| Fathers feel left out if a mother breastfeeds ** | 45.5 | 8.0 | 19.6 | 8.6 | 14.2 | 3.7±1.5 |
| Breast milk is the ideal food for babies | 1.7 | 0.0 | 1.3 | 2.3 | 94.3 | 3.9±0.6 |
| Breast milk is more easily digested than formula | 2.0 | 1.3 | 4.3 | 2.7 | 89.0 | 4.7±0.8 |
| Formula is as healthy for an infant as breast milk ** | 73.1 | 9.3 | 12.6 | 2.7 | 2.3 | 4.5±01.0 |
| Breastfeeding is more convenient than formula feeding | 26.2 | 6.3 | 11.3 | 5.0 | 51.1 | 3.5±1.7 |
| Breast milk is less expensive than formula | 6.3 | 1.7 | 3.0 | 2.2 | 86.7 | 4.6±1.7 |
| Mothers who occasionally drinks alcohol should not breastfeed** | 7.0 | 1.7 | 3.3 | 2.7 | 85.4 | 1.4±1.1 |
| Overall mean score on attitude toward infant feeding = 4.0 ± 1.0 | | | | | | |
| Total (summed) attitude score for participants was 65±7.2 (out of 85) | | | | | | |

* 1 = Strongly disagree, 2 = Disagree, 3 = Neutral, 4 = Agree, 5 = Strongly agree

**Statement with reverse score

**Table 4. Relationship between sociodemographic characteristics, pregnancy and delivery outcomes and total attitudes score of mothers toward breastfeeding (N = 301).**

| Variable | Categories | N | % | Total Attitude Score [a] | P-value |
|---|---|---|---|---|---|
| **Age (mean = 29.67, sd =)** | <24 | 25 | 8 | 67.9 | 0.16[b] |
| | 24–30 | 105 | 35 | 64.9 | |
| | 31–35 | 103 | 34 | 64.3 | |
| | >35 | 68 | 23 | 65.0 | |
| **Parity** | 1 | 84 | 27.9 | 65.4 | |
| | 2–3 | 131 | 43.5 | 64.7 | 0.42[b] |
| | ≥4 | 86 | 28.6 | 66.5 | |
| **Level of Education** | Primary or lower | 10 | 3.3 | 60.6 | |
| | Secondary level | 61 | 20.3 | 64.0 | 0.12[b] |
| | University degree or above | 137 | 45.5 | 65.2 | |
| | Graduate degree | 93 | 30.9 | 65.7 | |
| **Working status** | Employed | 130 | 43.2 | 65.07 | 0.84[c] |
| | Unemployed | 171 | 56.8 | 64.9 | |
| **Monthly family income** (n = 179) | 0–300 JD | 32 | 17.9 | 62.5 | |
| | 301–450 JD | 79 | 44.1 | 64.6 | **0.05**[b] |
| | 451–650 JD | 53 | 29.6 | 65.5 | |
| | >651 JD | 15 | 8.4 | 68.1 | |
| **Pregnancy Complications** | No | 205 | 68.1 | 64.5 | **0.05**[c] |
| | Yes | 96 | 31.9 | 66.1 | |
| **Delivery Complications** | No | 271 | 90 | 64.6 | **0.01**[c] |
| | Yes | 30 | 10 | 68.0 | |
| **Gestational Age (mean = 41.4 week)** | Preterm | 78 | 26 | 66.4 | **0.04**[c] |
| | Full term | 222 | 74 | 64.5 | |
| **Mode of Delivery** | Vaginal birth | 132 | 43.9 | 64.5 | 0.41[b] |
| | Planned CS | 81 | 26.9 | 64.9 | |
| | Emergency CS | 88 | 29.2 | 65.8 | |
| **Admission to NICU** | No admission to NICU | 247 | 82.0 | 64.8 | 0.36[b] |
| | Admission to NICU | 54 | 18.0 | 65.6 | |
| **Intention to Breastfeed** | Yes | 288 | 95.7 | 65.2 | **0.00**[b] |
| | No | 10 | 3.3 | 57.2 | |
| | Undetermined [d] | 3 | 1.0 | 65.3 | |
| **Willingness to Breastfeed Exclusively** | Yes | 106 | 35.2 | 66.7 | **0.01**[b] |
| | No | 146 | 48.5 | 63.8 | |
| | Undetermined | 49 | 16.3 | 64.9 | |
| **Breastfeeding counseling during antenatal care** | Yes/ prenatal | 60 | 20 | 64.9 | 0.99[b] |
| | Yes/ postnatal | 60 | 20 | 64.9 | |
| | No | 181 | 60 | 64.9 | |
| **Time of Initiation of Breastfeeding** | Day 1 | 130 | 43.2 | 65.3 | 0.51[b] |
| | Day 2 | 53 | 17.6 | 65.6 | |
| | Day 3 | 32 | 10.6 | 64.4 | |
| | Did not initiate | 86 | 28.6 | 64.0 | |

Mean for the sum of attitude score = 65.0 ± 7.2 out of 85 (range 37–81)

[a] Total attitude score is the sum of all items scores (range from 17 to 85). It was classified as positive to breastfeeding (a score of 70–85), neutral (a score of 49–69) and positive to formula feeding (a score of 17–48).

[b] One-way ANOVA

[c] Independent Sample t-test

[d] Undetermined refers to the case when the answer is neither yes nor no.

**Table 5. Determinants of positive attitude of mothers toward breastfeeding, results of logistic regression of family income, gestational age, intention to breastfeed, willingness to breastfeed exclusively, pregnancy complication, and delivery complications (N = 301)[a].**

| Variable | Categories | p-value | OR | 95%CI[b] |
|---|---|---|---|---|
| **Monthly family income** (n = 179) | 0–300 JD (reference) | | | |
| | 301–450 JD | **0.05[c]** | 4.88 | 1.12–23.8 |
| | 451–650 JD | 0.30 | 2.42 | 0.59–12.81 |
| | >651 JD | **0.00[c]** | 14.77 | 2.25–99.64 |
| **Gestational Age** | Preterm (reference) | | | |
| | Full term | 0.27 | 0.60 | 0.25–1.48 |
| **Pregnancy Complications** | No (reference) | | | |
| | Yes | 0.51 | 0.75 | 0.32–1.77 |
| **Delivery Complications** | No (reference) | | | |
| | Yes | 0.26 | 0.52 | 0.16–1.64 |
| **Intention to Breastfeed** | No (reference) | | | |
| | Yes | 0.98 | 1.04 | 0.25–1.49 |
| | Undetermined | 0.99 | 1.05 | 0.22–1.56 |
| **Willingness to Breastfeed Exclusively** | No (reference) | | | |
| | Yes | **0.01[c]** | 3.41 | 1.35–8.63 |
| | Undetermined | 0.91 | 1.09 | 0.36–3.31 |

[a] Model's Chi-square $(df)$ = 21.7 $(9)$, $p$ = .01, Nagelkerke $R^2$ = 28.4.

[b] Confidence Interval

[c] Statistically significant, $p < .05$

milk is an advanced combination of nutrients and that it is as good as breastmilk. This proportion is high compared to 9.2% having positive attitude to breastfeeding in Lebanon and Qatar [19], but considerably lower than 72% having positive attitude in Jordan in 2020 [29], and 55% in India [44]. Khasawneh et al (2020) have overestimated the proportion of mothers with positive attitude to breastfeeding. They offered the choice of only agree or disagree for the IIFAS scale, and agreement with more than half of the items was considered as positive attitude to breastfeeding.

Despite the fact that this hospital is a baby friendly hospital, only 43% of mothers initiated breastfeeding in the first day after childbirth, and 60% of mothers did not receive any counselling of advice related to breastfeeding. This may be explained by the hospital being a tertiary referral one, receiving a large volume of complicated cases requiring cesarean section, and inducing delay in initiation of breastfeeding. On the other hand, the role of this hospital in promoting positive breastfeeding behavior is suboptimal. It seems that many factors influence weather mothers' intentions are reflected into actual practice.

The mean total attitude score in this study was 65.0 ± 7.2, which is very similar to the mean total IIFAS score (63.5 ± 4.67) reported previously by Shoosha et.al. among Jordanian mothers [23]. Yet, this score is low compared to the very positive attitude with a total score of 81.39 ± 8.35 in Jeddah, Saudi Arabia [34], and to 69.76 ± 7.75 in Spain. On the other hand, lower total scores were reported in Poland (63.12 ± 7.34), in Saudi Arabia (59.6 ± 7.3) [35], and neutral attitude to breastfeeding (30 out of 50) in Fiji [9]. This indicates the multifaceted nature of the attitude toward breastfeeding, involving a wide array of social norms, culture and health care environment factors.

Findings of this study pointed out multiple sociodemographic characteristics and delivery characteristics that were associated with mother's attitudes to infant feeding. In accordance with other studies, socioeconomic status and particularly higher income, was associated with

more positive attitude to breastfeeding in Saudi Arabia [34], Fiji [9], Lebanon and Qatar [19]. In fact, higher family income continued to be a predictor of more positive attitude to breastfeeding, even after controlling for all other variables. Mothers with higher income level seem to care more about the quality of food they provide for their families, including choosing breastfeeding for their infants. Mothers with lowest income level demonstrated more positive attitude to formula milk feeding. These women cannot afford the price of formula milk, and have no choice but to breastfeed. This may explain why they favor formula milk more, and may perceive it as an ideal nutritious food for their infants.

On the other hand, mothers with higher education did not show more positive attitude to breastfeeding. This was consistent with results of other studies in the literature, where higher education was significantly associated with less positive attitude to breastfeeding, or had no association [9, 19, 29, 34, 35]. This may be explained by the fact that mothers with higher education are working or planning to work. For these mothers, bottle-feeding may be considered a more convenient option for infant feeding, since it is less time consuming and can be provided by any caregiver. These mothers may also value freedom, and prefer to choose infant feeding method that will not restrict their daily activities.

Interestingly, mothers who had pregnancy and delivery complications did not have significantly more positive attitude to breastfeeding. Similarly, positive attitude to breastfeeding was not associated with preterm delivery in this study. This comes in line with earlier findings in Jordan, where mothers of preterm infants were less likely to have positive attitude to breastfeeding [23]. On the other hand, other studies reported no association Saudi Arabia, Qater, Lebanon and in Jordan [19, 29, 34]. In addition, our results indicated that NICU admission did not influence postpartum attitude to breastfeeding as expected, which implies that NICU of the baby was not a barrier to breastfeeding among mothers in this study. Although admission to NICU was associated with a significant delay in the initiation of breastfeeding to second or third day after childbirth, it did not prevent initiation. This may be because this hospital is baby-friendly, and the healthcare providers do encourage and support mothers to initiate breastfeeding after birth.

Despite the fact that no association was found in this study between working status or employment and attitude to breastfeeding, other studies reported a negative association, where employment was a barrier to breastfeeding Saudi Arabia, Bangladesh, Qatar, Lebanon, and in Jordan [8, 14, 19, 29, 34, 35, 39, 45]. According to Al Tamimi et.al., 30% of the mothers in their study in Jordan, attributed premature cessation of breastfeeding to work [39]. In addition, Khasawneh et.al. indicated the short maternity leave as a prominent reason for non-exclusive feeding in Jordan [45]. The majority of mothers in the current study believed that mothers who were going back to their employment after childbirth, should choose formula feeding. Nevertheless, there was no difference in the mean attitude score between mothers who work and those who do not work, nor in their time of initiation of breastfeeding before discharge from hospital. This may suggest the more or less breastfeeding-friendly behavior that mothers perceive in many workplaces in Jordan.

We did not find any difference between mothers who received prenatal or postnatal counselling about breastfeeding and those who did not receive any counselling. In addition, no association was found between attitude and early initiation of breastfeeding. This is in line with findings of Khasawneh et.al. where counselling about breastfeeding by healthcare providers did not influence breastfeeding practice [29]. This may suggest that education and providing information about breastfeeding during pregnancy or just after delivery is not enough to influence mothers' attitude to breastfeeding.

This study demonstrated that mothers who were willing to breastfeed exclusively had more positive attitude to breastfeeding. This contradicts the findings from Lebanon, Qatar, Saudi

Arabia and from Jordan [8, 19, 29], where associations were reported between planning and intending to breastfeed.

To shape attitude to breastfeeding in mothers, it is essential to build more positive attitude to breastfeeding among husbands, mothers, mothers in law and friends of pregnant women [46]. Friends and family, among others in the community have a critical role in supporting, assisting, encouraging and motivating mothers to initiate and continue breastfeeding after childbirth. Their own attitudes to breastfeeding positively influence mothers' breastfeeding decisions and practices. They can effectively help mothers conquer barriers that may arise along the breastfeeding journey.

This study is the first to explore postnatal mothers' attitudes to breastfeeding in Jordan in addition to the influence of validated pregnancy and delivery outcomes on this attitude. Results of this study added insight to the evolving body of literature about mothers' breastfeeding attitudes in Jordan.

### Limitations

Nevertheless, there are some limitations of this study. First, the main challenge was that data collection started just before Covid-19 pandemic, which interrupted and delayed the process of data collection. There was the lock down initially, followed by limitation to admissions to the wards during that period. Additionally, temporary closure of the ward when cases were identified limited access of the research team for data collection. Thus, the sample size is less than planned beforehand. Second, results may not be generalizable to mothers in Jordan, since our sample did not include mothers with no health insurance who gave birth in private hospitals. In addition, the study was carried out in a teaching hospital, which is certified as a baby-friendly hospital. Therefore, results may have restricted generalizability to the population of mothers in Jordan who receive services in non-baby-friendly facilities. Furthermore, the sample was selected from mothers attending this referral teaching hospital in the capital Amman. The sample was selected from mothers attending this referral teaching hospital in the capital Amman. Many of women who receive service in this hospital are university employees and their families. This explains the high level of education among the participants. These women may not be representative of all women in Jordan.

Third, the interviewer-administered way of data collection for the attitude, may have resulted in social desirability bias in responses, despite the training provided for the interviewers to reduce this bias. Additionally, cultural beliefs about breastfeeding were not addressed as factors that may influence women's attitude toward breastfeeding in Jordan, which may be the focus of future research. Finally, data about initiation of breastfeeding in the first hour after birth is not available, which makes the data less useful for calculating the rate of early initiation of breastfeeding.

### Recommendations

Future research may explore women's attitudes before pregnancy, during pregnancy, and after delivery in order to investigate the effect of pregnancy and delivery experience on breastfeeding attitude. We also recommend more research targeting knowledge about breastfeeding. More qualitative studies can be of great benefit in exploring breastfeeding attitudes in the population in Jordan. We suggest that future studies look at whether the IIFAS is associated with breastfeeding practices in Jordan, controlling for other factors such as hospital breastfeeding routines (early initiation of breastfeeding, rooming-in, etc.), newborn characteristics (prematurity, birth weight, birth by cesarean section, etc.) and maternal characteristics (educational level, work status, income, etc.). We also recommend that future studies be conducted on a representative population sample with a prospective design so that results can contribute to policy recommendations for the Jordanian population.

There is an immediate need for interventions in Jordan to improve the rate of breastfeeding. Public policies can be implemented to increase the availability of designated breastfeeding places in workplaces, public places, parks, shopping malls, restaurants and health care facilities that are breastfeeding-friendly, prepared with a suitable place for breastfeeding. Policies and laws on marketing formula milk and breastmilk substitutes, paid maternity leave, availability of day care facilities, breastfeeding breaks for employed mothers, should all be enforced. Workplaces should encourage mothers to breastfeed their infants, and even reward them for adhering to this practice. Since the majority of women in Jordan give birth in private hospitals, more private hospitals better be encouraged to join the Baby Friendly Hospitals initiative. In addition, hospitals that are designated as baby friendly, must audit their procedures related to encouraging breastfeeding practice, and make sure that related policies are properly implemented. Interventions are specially needed to positively impact attitude to breastfeeding in low income communities, targeting women, their husbands, families and friends. Their support and encouragement is necessary for mothers to start and continue breastfeeding.

Health care students in all disciplines must be educated on breastfeeding. In addition, public education is essential to support and encourage woman's decision to breastfeed. To achieve that, initiatives must clearly communicate the benefits and importance of breastfeeding to the people [47], against the strong marketing for breastmilk substitutes by the companies.

Well-designed interventions for improving breastfeeding counselling by health care providers after birth is vital. Maternity nurses can be very effective in educating, supporting and encouraging women to breastfeed for the benefit of both the infant and the mother. Policies and laws on marketing formula milk and breastmilk substitutes, paid maternity leave, availability of day care facilities, breastfeeding breaks for employed mothers, should all be enforced. Workplaces should encourage mothers to breastfeed their infants, and even reward them for adhering to this practice.

## Conclusion

To conclude, this study found that the mothers in Jordan have neutral (attitude was close to the upper limit of the neutral range) attitude toward breastfeeding. In addition, it revealed the need for adopting various strategies for stimulating more positive attitude to breastfeeding, and especially people with low income, to optimize breastfeeding rates in Jordan. Healthcare professionals at all levels of the health care system must play their vital role in promoting breastfeeding, starting with prenatal counselling, and ending with immediate postnatal counselling and supporting mothers to initiate breastfeeding their infants. All this in the attempt to increase to the rates of breastfeeding among women in Jordan. Findings of this study can be instrumental for health care providers and policy makers in Jordan in their intentional efforts to boost breastfeeding on the population level.

## Supporting information

**S1 Dataset.**
(SAV)

## Acknowledgments

We would like to acknowledge Salameh Halaseh, Murad Haddad and Dima Nasrieh for data entry, management and cleaning. Also, Bdour Abdallat, Layan Ayesh, and Donia Jaber for their valuable efforts in data collection.

## Author Contributions

**Conceptualization:** Sireen M. Alkhaldi, Oqba Al-Kuran, Mai M. AlAdwan, Tala A. Dabbah, Heyam F. Dalky, Eiman Badran.

**Data curation:** Sireen M. Alkhaldi, Mai M. AlAdwan, Tala A. Dabbah.

**Formal analysis:** Sireen M. Alkhaldi.

**Investigation:** Sireen M. Alkhaldi, Oqba Al-Kuran, Heyam F. Dalky, Eiman Badran.

**Methodology:** Sireen M. Alkhaldi, Oqba Al-Kuran, Heyam F. Dalky, Eiman Badran.

**Project administration:** Sireen M. Alkhaldi, Oqba Al-Kuran, Mai M. AlAdwan, Tala A. Dabbah.

**Supervision:** Sireen M. Alkhaldi, Eiman Badran.

**Writing – original draft:** Sireen M. Alkhaldi, Mai M. AlAdwan, Tala A. Dabbah, Heyam F. Dalky.

**Writing – review & editing:** Sireen M. Alkhaldi, Oqba Al-Kuran, Mai M. AlAdwan, Tala A. Dabbah, Heyam F. Dalky, Eiman Badran.

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
