## [Decision Letter · Decision Letter 0]

31 Jan 2023

PONE-D-22-35675Determinants of Breastfeeding Attitudes of Mothers in Jordan: A Cross-sectional StudyPLOS ONE

Dear Dr. Alkhaldi,

Thank you for submitting your manuscript to PLOS ONE. After careful consideration, we feel that it has merit but does not fully meet PLOS ONE’s publication criteria as it currently stands. Therefore, we invite you to submit a revised version of the manuscript that addresses the points raised during the review process.

We look forward to receiving your revised manuscript.

Kind regards,

Ghada Abdrabo Abdellatif Elshaarawy, M.D

Academic Editor

PLOS ONE

Additional Editor Comments (if provided):

1) ABSTRACT:

No need for details of methodology in the abstract. Only main points. It should not include the type of statistical analysis.

Methodology has to mention the elements included as measuring parameter but without details.

2) INTRODUCTION:

Newest Global/ Regional/ Jordan prevalence of mothers’ attitude toward breastfeeding should be stated.

The current situation of other developed and developing countries should also be added.

The benefits of conducting the study to the community should be explained.

3) METHODS:

The characteristics of the study participants should be mentioned as inclusion criteria and exclusion criteria (if any).

How long did it take to complete each questionnaire?

What types of data (quantitative and/or qualitative) were used?

Definitions as postnatal period, Infant, exclusive breastfeeding, ……. etc. should be mentioned.

How did the authors get the study subjects? They have to clearly address their sampling technique?

Authors should include a reference for using the stated formula in calculating the sample size. Furthermore, the basis of sample size calculation should be mentioned to know the confidence level and the margin of error. Please give a justification for the sample size. How did you determine the sample size?

4) RESULTS:

In table 3, the sentence: “Overall mean attitude toward infant feeding was 3.95 ± 1.04 (out of 5)” need to be explained.

5) DISCUSSION:

Discuss by using the scientific reasoning the mothers’ attitude toward breastfeeding and the determinants of this attitude in other developing and developed countries with similar context. The manuscript could be greatly strengthened if the authors could compare the findings of the study with other findings and state the reasons for the strengths and weaknesses in each section.

6) CONCLUSION:

It should be specific and based on the findings of the study.

Reviewers' comments:

Reviewer's Responses to Questions

**Comments to the Author**

1. Is the manuscript technically sound, and do the data support the conclusions?

Reviewer #1: Yes

Reviewer #2: Partly

Reviewer #3: Yes

2. Has the statistical analysis been performed appropriately and rigorously? 

Reviewer #1: Yes

Reviewer #2: No

Reviewer #3: Yes

3. Have the authors made all data underlying the findings in their manuscript fully available?

Reviewer #1: Yes

Reviewer #2: No

Reviewer #3: Yes

4. Is the manuscript presented in an intelligible fashion and written in standard English?

Reviewer #1: Yes

Reviewer #2: No

Reviewer #3: Yes

5. Review Comments to the Author

Reviewer #1: Response to reviewers

The authors have submitted a manuscript titled “Determinants of breastfeeding attitudes of mothers in Jordan” This study is very useful and adds a significant value in the maternal and child health system of Ethiopia regarding the perception and attitudes of breastfeeding. However, I will recommend some of the minor suggestions to make this manuscript better.

1)In addition, exclusive breast feeding for six months, continued for two years and beyond with the provision of safe and appropriate complementary foods are the most important strategies for promoting child survival and health [4]. However, the benefits of breastfeeding are underestimated, and the proportion of mothers who breastfeed their new-borns is way below the recommended levels globally and Jordan is not an exception [5, 6].

In this statement Although importance of breastfeeding has been explained quite evidently, addition of relevant information in context of exclusive breastfeeding globally as well as regionally should be explained

2) These factors include mothers’ age, income, education, mode of delivery and availability of breastfeeding counselling before, during and after delivery, and past breastfeeding experience [7-13].

How the mentioned socio-demographic factors are crucial for breastfeeding experience should be elaborated. For ex- how education is important for prevalence of exclusive breastfeeding.

3) In addition, health professionals’ attitudes, type of health care facility where delivery occurred, and admission of the infant in neonatal intensive care unit (NICU) influence the process of breastfeeding by women [9, 14-17

Justification of how healthcare professionals’ attitude or type of care received is important for perception of breastfeeding

4) The target population for the current study was postnatal women in the maternity ward. The sample size calculated using a confidence interval of 95% and a margin of error of 5%. A sample size of atleast 290 was found to be adequate

How did the authors have selected the samples and how they were found to be adequate have they taken any reference from similar studies of same population or whether they have done any sample size calculation.

5) Friendly, prepared with a suitable place for breastfeeding. Policies and laws on marketing formula milk and breastmilk substitutes, paid maternity leave, availability of day care facilities, breastfeeding breaks for employed mothers, should all be enforced

In above statement Information regarding milk bank for those mothers who are unable to breastfeed the child due to less milk production.

6) Special counselling provided to mothers who had caesarean delivery and who have new-borns in the NICU is essential. Policies and laws on marketing formula milk and breastmilk substitutes, paid maternity leave, availability of day care facilities.

Importance of propagation of importance of breastfeeding through mhealth.,

Reviewer #2: This is interesting research. The authors need a little help in improving the writing and need to get a statistician on-board to help with the statistical test and appropriate interpretation. After making some improvements, the manuscript can get to a point where it is publishable in this journal.

Abstract

The first two statements in the abstract are not objective. They are also not presented in a scientific manner, but rather as a more casual and social statement. For the analysis, the authors use a multiple linear regression model to assess the association between attitude scores from the IIFAS questionnaire, and various maternal sociodemographic and pregnancy related characteristics. However, they report odds ratios, which is not the metric that results from a linear regression model. The authors need to get a statistician on their team.

Introduction:

End of paragraph 2:

Non-scientific expression, and no references – “The proportion of mothers who breastfeed their newborns ‘is way below’ the recommended levels globally, and Jordan ‘is not an’ exception”. (correct usage: Jordan is no exception).

End of paragraph 3:

Incomplete sentence – “Furthermore, several studies have demonstrated that mothers’ attitudes toward breastfeeding is a significant predictor of their intention, initiation, and continuation of.”

Paragraph 4 and 5 needs to be tighter and better worded for a scientific journal. For example, ‘What determines women’s attitude towards breastfeeding?’ can be stated as ‘We sought to assess factors that determine women’s attitude towards breastfeeding’. Also, the authors do not build interest on the need for their research in an effective manner.

Methods:

Study Design and setting

COVID lockdown and responsive adjustments at the study setting looks like an unnecessary sub-heading. These details can just be incorporated into the paragraph on setting.

Population and Sample

Sample size calculation: The Section on Population and Sample is too concise and needs further detail with respect to power and sample size calculations. Insufficient details are given, and the sentence ‘A sample size of at least 290 was found to be adequate’ is lacking. The authors need to explain what a sample size of 290 is adequate for. This section is a good place to discuss sample size and power calculations. What is the minimal difference that can be detected with a sample size of 290?

Study Instrument

This section is not presented with proper English grammar, punctuation, and sentence structure. It needs to be presented better. Some sentences in this section appear like bullet points instead of complete sentences. For example, “Iowa Infant Feeding Attitude Scale, a validated tool used to measure attitudes of mothers toward infant feeding.”. Other sentences are too long and need to be broken down. For example, “Eight statements indicate positive attitude toward breastfeeding and nine statements indicate positive attitude towards formula feeding, the latter were reverse-scored before calculating the total scores of all items”. Other statements that are grammatically incorrect include “The researcher used a validated Arabic version of the IIFAS was used after permission from the author”.

The second paragraph in this section includes a sentence that starts with “The fourth aspect”. This is confusing, and the authors did not explicitly indicate what the first three aspects were. Also, the same sentence continues on to read ‘abased’ instead of ‘based’.

Data collection procedure

The first sentence has incorrect punctuation, lacks proper structure, and grammar.

Correct grammar would imply using ‘completing’ instead of ‘completed’ in the statement: “The response rate was 95%, with 301 mothers ‘completing’… questions”.

Analysis

This section mentions, “To determine the specific factors that influence breastfeeding attitude among participating mothers in the current study, independent t-test and one-way ANOVA statistical tests were performed”. However, the results as presented in Table 4 can be more appropriately stated as follows: “Independent t-test and one-way ANOVA tests were performed to assess if the total attitude score towards breastfeeding different significantly across each of the measured patient characteristics”.

The section mentions the use of ‘multiple linear regression models’ but talks about expressing results as adjusted odds ratios with 95% CI. Logistic regression models produce odds ratios, and the statement used in this section is not congruent with the results produced in the subsequent section.

Results

It is more appropriate to use the term ‘presents’ than ‘demonstrates’ when talking about the tables. For example, ‘Table 1 presents sociodemographic characteristics…. information”.

The sentence is not correctly constructed: “The most common pregnancy complication among mothers was gestational diabetes (16%), and 26% had preterm deliveries constitutes 26% of infants”.

The title for table 1 does not make sense.

The authors need to be consistent in the tables and in the text with regards to decimal places on percentages (e.g., use one decimal place throughout the manuscript for %s) and make sure the percentages reported in the table and the manuscript text match exactly.

The title for table 2 needs to be improved.

The sentence right above Table 2 reads ‘firth’ instead of ‘birth’.

The IIFAS statement with the second highest mean score as mentioned in the text in the paragraph above table 3 does not match with what is presented in Table 3.

The statement “This indicates a neutral postnatal attitude to breastfeeding” is unsubstantiated based on a mean total summed attitude score of 65 and a standard deviation of 7.15. There needs to be a statistical test that proves this statement.

In the paragraph that starts with the statement ‘Table 4 displays’, the second sentence mentions ‘attitude towards breastfeeding was significantly higher’, but a higher attitude is not a valid statement. Perhaps a ‘higher score’ or a ‘more positive attitude’ would be a more suitable word.

In the next paragraph right after Table 4, the authors list 6 independent variables, but the sentence reads, “Of the five independent variables”.

The results as presented in Table 5 need to be presented better and the title for Table 5 needs to be more concise. It appears that the authors used a multiple linear regression model, with the participant’s ‘Total attitude score’ as the dependent variable, and age, education, income, intention to breastfeed, gestational age, and neonatal ICU admission as the independent variables. However, at least four of the independent variables would be categorical variables. The author needs to state which category was used as the reference, particularly for categories with more than two response options, example, Education. Furthermore, the authors do not interpret these coefficients in the results section, either by their unstandardized estimates, or standardized estimates. It must be pointed out that manuscripts don’t typically report standardized coefficients from a multiple linear regression model since they are not directly interpretable. Unstandardized coefficients use the natural unit of measure for the specific variable and can be easily interpreted, and are therefore, the only reported coefficients in scientific manuscripts. For example, based on the results in Table 5, a unit increase in income results in a 1.61-point increase in the attitude score. However, this interpretation would only apply if income was measured as a continuous variable. However, the authors did not state if income is a continuous variable in their model, and in case it was categorical, the reference category is not mentioned. This makes interpreting the coefficients difficult. Assuming that the model is specified correctly, the p-values would suggest the variables/maternal factors that are significantly associated with the total scores on the IIFAS questionnaire. Finally, it appears that the authors are interpreting the standardized coefficients as odds ratios based on their statement in the abstract. This is incorrect, and a statistician would be a good addition to the team in improving this paper.

Reviewer #3: Determinants of Breastfeeding Attitudes of Mothers in Jordan: A Cross-sectional Study

Revision of the study “Determinants of Breastfeeding Attitudes of Mothers in Jordan: A Cross-sectional Study”.

This study aims to investigate the Breastfeeding Attitudes of Mothers in Jordan; during the study period January 2020 to May 2021.

Evidence suggests that the breastfeeding attitudes are higher in low-income mothers and general people.

Limitations of the study are noted and discussed.

I think the paper covers an import area of research.

I have listed some specific comments below that the authors should take into account before this

work could be published.

Abstract:

“The mean attitude score for participants was 65.0 ±7.15, indicating a neutral attitude. Factors associated with positive breastfeeding attitude were high income (p=0.048), pregnancy complications (p= 0.049), delivery complications (p= 0.008), prematurity (p= 0.042), intention to breastfeed (p= 0.002) and willingness to breastfeed (p= 0.005). Multiple linear regression results indicated higher income and intention to breastfeed as predictors of positive breastfeeding attitude (AOR 0.019; 95% CI 0.30-2.91 and AOR 1.52; 95%CI 0.05-10.20 respectively).”

I believe focusing on these risk factors should be the main aim of the study. What do you

mean when you say: “positive breastfeeding attitude”? Please specify if these were protective or risk factors.

Introduction:

1. Could you please add 2 or 3 references about the global scenario regarding Breastfeeding Attitudes of Mothers?

Method:

1. Which sampling technique authors were used?

Result:

The tables and figures are well described.

Discussion:

Overall discussion was well written. Just add 2 or 3 references which is related to your significant results.

6. PLOS authors have the option to publish the peer review history of their article (what does this mean?). If published, this will include your full peer review and any attached files.

Reviewer #1: **Yes: **Dr Ritik Agrawal

Reviewer #2: No

Reviewer #3: **Yes: **Sadia Afrin

---

## [Author Response · Author response to Decision Letter 0]

11 Feb 2023

PONE-D-22-35675

Determinants of Breastfeeding Attitudes of Mothers in Jordan: A Cross-sectional Study

PLOS ONE

Author responses are marked in BLUE

Additional Editor Comments (if provided):

1) ABSTRACT:

No need for details of methodology in the abstract. Only main points. It should not include the type of statistical analysis. Methodology has to mention the elements included as measuring parameter but without details.

Names of statistical tests have been removed. 



2) INTRODUCTION:

Newest Global/ Regional/ Jordan prevalence of mothers’ attitude toward breastfeeding should be stated.

I have added and included what indicates the state of attitude to breastfeeding globally, regoionally, and in Jordan.

The current situation of other developed and developing countries should also be added.

The previous point included developed and developing countries.

The benefits of conducting the study to the community should be explained.

Statements have been added to indicate benefits to the community, such as: “Advancing better understanding of the country-specific determinants and predictors of women’s attitudes toward breastfeeding is critical for development of effective health care programs, and for informing health care providers and policy makers in Jordan.”

3) METHODS:

The characteristics of the study participants should be mentioned as inclusion criteria and exclusion criteria (if any).

There were no inclusion or exclusion criteria

How long did it take to complete each questionnaire?

The answer was added: it took 10 minutes to answer each questionnaire. 

What types of data (quantitative and/or qualitative) were used?

Answer provided: all data were quantitative .

Definitions as postnatal period, Infant, exclusive breastfeeding, ……. etc. should be mentioned.

Definitions were provided in the appropriate place in the methods.

How did the authors get the study subjects? They have to clearly address their sampling technique?

Details were provided.

Authors should include a reference for using the stated formula in calculating the sample size. Furthermore, the basis of sample size calculation should be mentioned to know the confidence level and the margin of error. Please give a justification for the sample size. How did you determine the sample size?

A reference was provided for the sample size calculator used. 

4) RESULTS:

In table 3, the sentence: “Overall mean attitude toward infant feeding was 3.95 ± 1.04 (out of 5)” need to be explained.

(out of 5) was deleted. No need for it since it is made clear in the table that 5-point Likert was used.

5) DISCUSSION:

Discuss by using the scientific reasoning the mothers’ attitude toward breastfeeding and the determinants of this attitude in other developing and developed countries with similar context. The manuscript could be greatly strengthened if the authors could compare the findings of the study with other findings and state the reasons for the strengths and weaknesses in each section.

Comparisons with developed and developing countries has bees introduced.

6) CONCLUSION:

It should be specific and based on the findings of the study.

I have deletes sentences not based on results of the study.

[

Reviewer #1: Response to reviewers

The authors have submitted a manuscript titled “Determinants of breastfeeding attitudes of mothers in Jordan” This study is very useful and adds a significant value in the maternal and child health system of Ethiopia regarding the perception and attitudes of breastfeeding. However, I will recommend some of the minor suggestions to make this manuscript better.

1)In addition, exclusive breast feeding for six months, continued for two years and beyond with the provision of safe and appropriate complementary foods are the most important strategies for promoting child survival and health [4]. However, the benefits of breastfeeding are underestimated, and the proportion of mothers who breastfeed their new-borns is way below the recommended levels globally and Jordan is not an exception [5, 6].

In this statement Although importance of breastfeeding has been explained quite evidently, addition of relevant information in context of exclusive breastfeeding globally as well as regionally should be explained

Breastfeeding tares in Jordan and in the region have been added.

2) These factors include mothers’ age, income, education, mode of delivery and availability of breastfeeding counselling before, during and after delivery, and past breastfeeding experience [7-13].

How the mentioned socio-demographic factors are crucial for breastfeeding experience should be elaborated. For ex- how education is important for prevalence of exclusive breastfeeding.

More explanation has been added about the positive or negative effect of certain factors on breastfeeding attitudes.

3) In addition, health professionals’ attitudes, type of health care facility where delivery occurred, and admission of the infant in neonatal intensive care unit (NICU) influence the process of breastfeeding by women [9, 14-17

Justification of how healthcare professionals’ attitude or type of care received is important for perception of breastfeeding

Explanation was provided.

4) The target population for the current study was postnatal women in the maternity ward. The sample size calculated using a confidence interval of 95% and a margin of error of 5%. A sample size of atleast 290 was found to be adequate

How did the authors have selected the samples and how they were found to be adequate have they taken any reference from similar studies of same population or whether they have done any sample size calculation.

I have added explanation and reference for the sample size calculation, and provided rationale for the sample selection and size. 

5) Friendly, prepared with a suitable place for breastfeeding. Policies and laws on marketing formula milk and breastmilk substitutes, paid maternity leave, availability of day care facilities, breastfeeding breaks for employed mothers, should all be enforced.

In above statement Information regarding milk bank for those mothers who are unable to breastfeed the child due to less milk production.

The idea of a “milk bank” may not be appropriate to propose in a country like Jordan, due to cultural and religious issues.

6) Special counselling provided to mothers who had caesarean delivery and who have new-borns in the NICU is essential. Policies and laws on marketing formula milk and breastmilk substitutes, paid maternity leave, availability of day care facilities.

Importance of propagation of importance of breastfeeding through mhealth.,

I have added the concept of importance of breastfeeding through maternal health services.

Reviewer #2: 

This is interesting research. The authors need a little help in improving the writing and need to get a statistician on-board to help with the statistical test and appropriate interpretation. After making some improvements, the manuscript can get to a point where it is publishable in this journal.

I would like to thank Reviewer #2 for the very constructive comments and suggestions for improving the manuscript. 

Abstract

The first two statements in the abstract are not objective. They are also not presented in a scientific manner, but rather as a more casual and social statement.

First two sentences replaced by a more objective sentence.

 For the analysis, the authors use a multiple linear regression model to assess the association between attitude scores from the IIFAS questionnaire, and various maternal sociodemographic and pregnancy related characteristics. However, they report odds ratios, which is not the metric that results from a linear regression model. The authors need to get a statistician on their team.

That was an evident mistake. Any way, a statistician has been consulted and I have replaced multiple linear regression with logistic regression. 

Introduction:

End of paragraph 2:

Non-scientific expression, and no references – “The proportion of mothers who breastfeed their newborns ‘is way below’ the recommended levels globally, and Jordan ‘is not an’ exception”. (correct usage: Jordan is no exception).

Valid references have been provided to indicate the low rates of breastfeeding globally. 

The expression has been corrected to become: Jordan is no exception. 

End of paragraph 3:

Incomplete sentence – “Furthermore, several studies have demonstrated that mothers’ attitudes toward breastfeeding is a significant predictor of their intention, initiation, and continuation of.”

True, sentence was incomplete. The missing word is added: breastfeeding.

Paragraph 4 and 5 needs to be tighter and better worded for a scientific journal. For example, ‘What determines women’s attitude towards breastfeeding?’ can be stated as ‘We sought to assess factors that determine women’s attitude towards breastfeeding’. Also, the authors do not build interest on the need for their research in an effective manner.

The question was deleted, and suggested sentence added. In addition to other changes to the paragraph. 

Methods:

Study Design and setting

COVID lockdown and responsive adjustments at the study setting looks like an unnecessary sub-heading. These details can just be incorporated into the paragraph on setting.

I agree, the subheading was deleted and details included in setting.

Population and Sample

Sample size calculation: The Section on Population and Sample is too concise and needs further detail with respect to power and sample size calculations. Insufficient details are given, and the sentence ‘A sample size of at least 290 was found to be adequate’ is lacking. The authors need to explain what a sample size of 290 is adequate for. This section is a good place to discuss sample size and power calculations. What is the minimal difference that can be detected with a sample size of 290?

A new calculation for sample size was provided with reference, which is more detailed.

Study Instrument

This section is not presented with proper English grammar, punctuation, and sentence structure. It needs to be presented better. Some sentences in this section appear like bullet points instead of complete sentences. For example, “Iowa Infant Feeding Attitude Scale, a validated tool used to measure attitudes of mothers toward infant feeding.”. 

The sentence has been completed; see track changes.

Other sentences are too long and need to be broken down. For example, “Eight statements indicate positive attitude toward breastfeeding and nine statements indicate positive attitude towards formula feeding, the latter were reverse-scored before calculating the total scores of all items”.

This very long sentence has been broken down into two sentences, and adjusted to be clearer. 

Other statements that are grammatically incorrect include “The researcher used a validated Arabic version of the IIFAS was used after permission from the author”.

The sentence was corrected. 

The second paragraph in this section includes a sentence that starts with “The fourth aspect”. This is confusing, and the authors did not explicitly indicate what the first three aspects were.

This has been correctd.

 Also, the same sentence continues on to read ‘abased’ instead of ‘based’.

Abased is incorrect, it was replaced with “based”.

Data collection procedure

The first sentence has incorrect punctuation, lacks proper structure, and grammar.

Sentence was rearranged and edited.

Correct grammar would imply using ‘completing’ instead of ‘completed’ in the statement: “The response rate was 95%, with 301 mothers ‘completing’… questions”.

This was corrected to “completing”

Analysis

This section mentions, “To determine the specific factors that influence breastfeeding attitude among participating mothers in the current study, independent t-test and one-way ANOVA statistical tests were performed”. However, the results as presented in Table 4 can be more appropriately stated as follows: “Independent t-test and one-way ANOVA tests were performed to assess if the total attitude score towards breastfeeding different significantly across each of the measured patient characteristics”.

I replaces the old sentence with the one reviewer #2 suggested. 

The section mentions the use of ‘multiple linear regression models’ but talks about expressing results as adjusted odds ratios with 95% CI. Logistic regression models produce odds ratios, and the statement used in this section is not congruent with the results produced in the subsequent section.

The comment was perfectly in place. I preferred to replace multiple linear regression with logistic regression, which was easier to interpret. Still. Results did not change, the same two variables stayed significant.

Results

It is more appropriate to use the term ‘presents’ than ‘demonstrates’ when talking about the tables. For example, ‘Table 1 presents sociodemographic characteristics…. information”.

Demonstrates was replaced with presents.

The sentence is not correctly constructed: “The most common pregnancy complication among mothers was gestational diabetes (16%), and 26% had preterm deliveries constitutes 26% of infants”.

The sentence was corrected: The most common pregnancy complication among mothers was gestational diabetes (16%), and 26% of mothers had preterm deliveries

The title for table 1 does not make sense.

Title has been corrected: Sociodemographic characteristics, pregnancy and delivery variables of mothers participating in this study about attitudes toward breastfeeding (N = 301)

The authors need to be consistent in the tables and in the text with regards to decimal places on percentages (e.g., use one decimal place throughout the manuscript for %s) and make sure the percentages reported in the table and the manuscript text match exactly.

The title for table 2 needs to be improved.

Title has been improved.: Breastfeeding- related variables of mothers participating in this study about attitudes toward breastfeeding (N = 301)

The sentence right above Table 2 reads ‘firth’ instead of ‘birth’.

The word was corrected: birth.

The IIFAS statement with the second highest mean score as mentioned in the text in the paragraph above table 3 does not match with what is presented in Table 3. 

The corrected item was placed.

The statement “This indicates a neutral postnatal attitude to breastfeeding” is unsubstantiated based on a mean total summed attitude score of 65 and a standard deviation of 7.15. There needs to be a statistical test that proves this statement.

In study instrument section of the Methods: total attitude score was classified as positive to breastfeeding (a score of 70 - 85), neutral (a score of 49 - 69) and positive to formula feeding (a score of 17-48). This classification is added as a foot note beneath the table. 

In the paragraph that starts with the statement ‘Table 4 displays’, the second sentence mentions ‘attitude towards breastfeeding was significantly higher’, but a higher attitude is not a valid statement. Perhaps a ‘higher score’ or a ‘more positive attitude’ would be a more suitable word.

This was corrected: higher total attitude score.

In the next paragraph right after Table 4, the authors list 6 independent variables, but the sentence reads, “Of the five independent variables”.

The paragraph was deleted, though the comment is valid.

The results as presented in Table 5 need to be presented better and the title for Table 5 needs to be more concise. It appears that the authors used a multiple linear regression model, with the participant’s ‘Total attitude score’ as the dependent variable, and age, education, income, intention to breastfeed, gestational age, and neonatal ICU admission as the independent variables. However, at least four of the independent variables would be categorical variables. The author needs to state which category was used as the reference, particularly for categories with more than two response options, example, Education. Furthermore, the authors do not interpret these coefficients in the results section, either by their unstandardized estimates, or standardized estimates. It must be pointed out that manuscripts don’t typically report standardized coefficients from a multiple linear regression model since they are not directly interpretable. Unstandardized coefficients use the natural unit of measure for the specific variable and can be easily interpreted, and are therefore, the only reported coefficients in scientific manuscripts. For example, based on the results in Table 5, a unit increase in income results in a 1.61-point increase in the attitude score. However, this interpretation would only apply if income was measured as a continuous variable. However, the authors did not state if income is a continuous variable in their model, and in case it was categorical, the reference category is not mentioned. This makes interpreting the coefficients difficult. Assuming that the model is specified correctly, the p-values would suggest the variables/maternal factors that are significantly associated with the total scores on the IIFAS questionnaire. 

It seems that I did faced some problems interpreting results of multiple linear regression. 

I have consulted statistician. I decided to replace multiple linear regression with logistic regression. 

Finally, it appears that the authors are interpreting the standardized coefficients as odds ratios based on their statement in the abstract. This is incorrect, and a statistician would be a good addition to the team in improving this paper.

That was a mistake actually. 

Reviewer #3: 

Determinants of Breastfeeding Attitudes of Mothers in Jordan: A Cross-sectional Study

Revision of the study “Determinants of Breastfeeding Attitudes of Mothers in Jordan: A Cross-sectional Study”.

This study aims to investigate the Breastfeeding Attitudes of Mothers in Jordan; during the study period January 2020 to May 2021.

Evidence suggests that the breastfeeding attitudes are higher in low-income mothers and general people.

Limitations of the study are noted and discussed.

I think the paper covers an import area of research.

I have listed some specific comments below that the authors should take into account before this

work could be published.

Abstract:

“The mean attitude score for participants was 65.0 ±7.15, indicating a neutral attitude. Factors associated with positive breastfeeding attitude were high income (p=0.048), pregnancy complications (p= 0.049), delivery complications (p= 0.008), prematurity (p= 0.042), intention to breastfeed (p= 0.002) and willingness to breastfeed (p= 0.005). Multiple linear regression results indicated higher income and intention to breastfeed as predictors of positive breastfeeding attitude (AOR 0.019; 95% CI 0.30-2.91 and AOR 1.52; 95%CI 0.05-10.20 respectively).”

I believe focusing on these risk factors should be the main aim of the study. What do you

mean when you say: “positive breastfeeding attitude”? 

“positive breastfeeding attitude” was corrected to become: Attitude is positive to breastfeeding”

Please specify if these were protective or risk factors.

Multiple linear regression was replaced with Logistic regression.

New analysis: 

With binary logistic regression modelling, predictors of positive attitude to breastfeeding were highest income and willingness to breastfeed exclusively (AOR= 18.97, 95%CI= 3.00-124.11 and AOR= 3.49, 95%CI= 0.36-3.31 respectively). Looking at the odds ratios, attitude that is positive to breastfeeding was significantly higher among mothers with higher income and mothers who were willing to breastfeed exclusively. 

Introduction:

1. Could you please add 2 or 3 references about the global scenario regarding Breastfeeding Attitudes of Mothers?

References have been added. 

Method:

1. Which sampling technique authors were used?

I have added the reference for the sample size calculator used. I have elaborated more on the way women were included in the study. 

Result:

The tables and figures are well described.

Titles of table 1 and table 2 have been revised.

Discussion:

Overall discussion was well written. Just add 2 or 3 references which is related to your significant results.

References have been added as recommended.

---

## [Decision Letter · Decision Letter 1]

12 Mar 2023

PONE-D-22-35675R1Determinants of Breastfeeding Attitudes of Mothers in Jordan: A Cross-sectional StudyPLOS ONE

Dear Dr. Alkhaldi,

Thank you for submitting your manuscript to PLOS ONE. After careful consideration, we feel that it has merit but does not fully meet PLOS ONE’s publication criteria as it currently stands. Therefore, we invite you to submit a revised version of the manuscript that addresses the points raised during the review process.

We look forward to receiving your revised manuscript.

Kind regards,

Ghada Abdrabo Abdellatif Elshaarawy, M.D

Academic Editor

PLOS ONE

Journal Requirements:

Additional Editor Comments (if provided):

Reviewer #2 comment which was not corrected:

The authors need to be consistent in the tables and in the text with regards to decimal places on percentages (e.g., use one decimal place throughout the manuscript for %s) and make sure the percentages reported in the table and the manuscript text match exactly.

Make sure you thoroughly address the reviewers' comments and concerns and ensure the manuscript is free of any editorial or grammatical errors.

Reviewers' comments:

Reviewer's Responses to Questions

**Comments to the Author**

1. If the authors have adequately addressed your comments raised in a previous round of review and you feel that this manuscript is now acceptable for publication, you may indicate that here to bypass the “Comments to the Author” section, enter your conflict of interest statement in the “Confidential to Editor” section, and submit your "Accept" recommendation.

Reviewer #4: All comments have been addressed

Reviewer #5: (No Response)

Reviewer #6: (No Response)

Reviewer #7: All comments have been addressed

2. Is the manuscript technically sound, and do the data support the conclusions?

Reviewer #4: Yes

Reviewer #5: Yes

Reviewer #6: Partly

Reviewer #7: Yes

3. Has the statistical analysis been performed appropriately and rigorously? 

Reviewer #4: Yes

Reviewer #5: Yes

Reviewer #6: I Don't Know

Reviewer #7: Yes

4. Have the authors made all data underlying the findings in their manuscript fully available?

Reviewer #4: No

Reviewer #5: Yes

Reviewer #6: (No Response)

Reviewer #7: Yes

5. Is the manuscript presented in an intelligible fashion and written in standard English?

Reviewer #4: Yes

Reviewer #5: No

Reviewer #6: No

Reviewer #7: Yes

6. Review Comments to the Author

Reviewer #4: The study involves analyzing data of breastfeeding attitudes of mothers in Jordan that gathered from one teaching hospital in Amman to examine the attitudes of mothers toward breastfeeding. The methods used a questionnaire according to Iowa Infant Feeding Attitude Scale to measure attitudes of mothers toward infant feeding. The results showed that the mean attitude score for participants was 65.0 ±7.15, indicating a neutral attitude. Factors associated with positive breastfeeding attitude were high income (p=0.048), pregnancy complications (p= 0.049), delivery complications (p= 0.008), prematurity (p= 0.042), intention to breastfeed (p= 0.002) and willingness to breastfeed (p= 0.005).

Comments and inquiries below are mostly minor: Please see the attached document:

1. Could you include how the participants were recruited? Did they receive hard copy, or they answered orally? Explain the method.

2. Was there a complete lockdown in the delivery section? If yes, mention the period.

3. Could you state the inclusion or exclusion criteria for selecting the participants?

4. Was there an informed consent or not?

5. Did you use any sampling method? Like random or systemic sampling technique for selecting the participants.

6. It looks like interview. Is that right? If yes, did all the patients answered verbally? How long was it take? If mixed, written or interview, what was the percentage of each. Did you record the interview?

7. In the data collection procedure section: line 9: Medical students entered the answers electronically. Do you mean for analyzing the data? Clarify.

8. Did you take the permissions from the patients before collecting these data? Explain

9. In the analysis section: In addition, the mean and standard deviation for individual items of IIFAS, and total score for the attitude toward breastfeeding for each respondent were calculated. How? Which test did you use?

10. In table 4: undetermined: Why, what was the reason? explain that in the footnote under the table.

11. Why are mothers more likely to use formula than breastfeed?

12. In the end of page 14: healthcare provider: Did they use ads for breastfeeding? What kind of encourage did they provide?

Reviewer #5: Comment” The characteristics of the study participants should be mentioned as inclusion criteria and exclusion. The authors responded “There were no inclusion or exclusion criteria” . It cannot be, there must be inclusion and exclusion criteria for every study and yours of no exception .

Other comments by reviewers were endorsed adequately by the authors.

Further Comments:

Under Introduction: the statement “Yet, About 44% of infants 0–6 months old are exclusively breastfed over the period of 2015-2020 [4]. Where? I cannot access the link for reference 4.

Under Introduction: the statement “The decision about infant feeding seems to be made before childbirth” the authors quoted reference [21], however this conclusion was not mentioned in that article. Please revise.

The objective of the study should be stated clearly at the end of the introduction.

What was the validity and reliability of IIFAS questionnaire

Item 17 of IIFAS questionnaire is about drinking alcohol for breastfeeding women. I find it difficult to ask Muslim Jordanian women this question. Please address my concern.

The authors stated that “ The researcher used a validated Arabic version of the IIFAS was used after permission from the author” and they quoted reference 29 for this statement. Indeed, the authors under reference 29 used a semi structured questionnaire survey and they did not use IIFAS questionnaire, please revise.

Under data collection procedure: the authors sated that “response rate was 95% with 301 mothers completing the questionnaires” What was the dominator for 301 mothers?

The authors stated that “monthly family income of 450 Jordan Dinar or less” please convert this to US$ as readers are unfamiliar with Jordan Dinar.

Reviewer #6: General comment:

Based on other studies on the determinants of breastfeeding, some of which are included in the manuscript bibliography, my question to the authors is: why was “knowledge of breastfeeding” not included in the study question? The Iowa Infant Feeding Attitude Survey (IIFAS) questionnaire used for data collection has items that can be better classified as “knowledge” rather than as “attitude”. The IIFAS tool developers could be asked as to whether they intentionally classified all survey items as assessment of “attitude” when in fact they appear to be testing “knowledge”.

Parturient mothers may be less intimidated to answer questions on their “attitude” and may be more apt to accept participation in the study, as opposed to thinking that they would be tested and evaluated with a survey of their correct or incorrect “knowledge,” whereas attitudes are neither right or wrong. However, the research needs to determine if the items are truly measuring “attitude” or “knowledge.” The manuscript would be clearer to readers if the authors could discuss the issue of “knowledge” versus “attitude” in relation to intent and practice of breastfeeding, and why they have excluded the assessment of knowledge. There may be studies available in the literature that address that issue.

A related issue is the possible presence of myths and cultural beliefs related to breastfeeding practice, and how those affect breastfeeding attitudes. These do not seem to have been addressed in the manuscript, though these myths and beliefs are highly prevalent in many societies. The authors should discuss the possible presence or not of breastfeeding myths and cultural beliefs in their study population of Jordan.

Specific comments:

Abstract.

Second sentence has a misspelling of “determine.”

Methodology

Medical students were used to interview mothers for this study. What percentage of these were female? There are two intersecting issues. One is that same-sex (female) interviewers may be more likely to obtain truthful answers from female interviewees. The second is that the interviewees may be also intimidated by perceived better-educated and perceived higher-status medical students, and may provide answers that mothers perceive as the “correct answers” rather than their true attitudes or knowledge.

Results

Table 3 should indicate that the items were taken from the IIFAS questionnaire with reference cited.

At the bottom of Table 3, the label should read “Overall mean score on attitudes toward infant feeding = 3.95 + 1.04”.

Readers may likely wonder why an attitude score of 3.95 + 1.04 is considered by the authors as a “neutral” attitude. This designation should be reconsidered.

Recommendations

Recommendations are primarily to do more studies on “attitudes.” My question again is related to the issue of breastfeeding “knowledge.” Are more studies needed on breastfeeding knowledge as well as on cultural beliefs and myths related to breastfeeding?

Reviewer #7: The paper is generally well written and structured. Overall, this is a clear, concise, and well-written manuscript. The introduction is relevant. Sufficient information about the previous study findings is presented for readers.

7. PLOS authors have the option to publish the peer review history of their article (what does this mean?). If published, this will include your full peer review and any attached files.

Reviewer #4: No

Reviewer #5: No

Reviewer #6: **Yes: **Laura Altobelli

Reviewer #7: **Yes: **Dr. Mahmoud Al-Masaeed

---

## [Author Response · Author response to Decision Letter 1]

15 Mar 2023

PONE-D-22-35675R1

Determinants of Breastfeeding Attitudes of Mothers in Jordan: A Cross-sectional Study

PLOS ONE

Journal Requirements:

References have been revised and corrected.

Additional Editor Comments (if provided):

Reviewer #2 comment which was not corrected:

The authors need to be consistent in the tables and in the text with regards to decimal places on percentages (e.g., use one decimal place throughout the manuscript for %s) and make sure the percentages reported in the table and the manuscript text match exactly.

I have made all numbers and percentages with one decimal point. Except p-values and confidence intervals, which I thought that two decimal points would be more appropriate. I have also referred to papers published in PLOS ONE and found p-vales and CI presented with two decimal points.

Make sure you thoroughly address the reviewers' comments and concerns and ensure the manuscript is free of any editorial or grammatical errors.

Reviewers' comments:

Reviewer's Responses to Questions

Comments to the Author

1. If the authors have adequately addressed your comments raised in a previous round of review and you feel that this manuscript is now acceptable for publication, you may indicate that here to bypass the “Comments to the Author” section, enter your conflict of interest statement in the “Confidential to Editor” section, and submit your "Accept" recommendation.

Reviewer #4: All comments have been addressed

Reviewer #5: (No Response)

Reviewer #6: (No Response)

Reviewer #7: All comments have been addressed

2. Is the manuscript technically sound, and do the data support the conclusions?

Reviewer #4: Yes

Reviewer #5: Yes

Reviewer #6: Partly

Reviewer #7: Yes

3. Has the statistical analysis been performed appropriately and rigorously?

Reviewer #4: Yes

Reviewer #5: Yes

Reviewer #6: I Don't Know

Reviewer #7: Yes

4. Have the authors made all data underlying the findings in their manuscript fully available?

Reviewer #4: No

Reviewer #5: Yes

Reviewer #6: (No Response)

Reviewer #7: Yes

I have made the data set available with the last revision.

5. Is the manuscript presented in an intelligible fashion and written in standard English?

Reviewer #4: Yes

Reviewer #5: No

Reviewer #6: No

Reviewer #7: Yes

6. Review Comments to the Author

Reviewer #4: The study involves analyzing data of breastfeeding attitudes of mothers in Jordan that gathered from one teaching hospital in Amman to examine the attitudes of mothers toward breastfeeding. The methods used a questionnaire according to Iowa Infant Feeding Attitude Scale to measure attitudes of mothers toward infant feeding. The results showed that the mean attitude score for participants was 65.0 ±7.15, indicating a neutral attitude. Factors associated with positive breastfeeding attitude were high income (p=0.048), pregnancy complications (p= 0.049), delivery complications (p= 0.008), prematurity (p= 0.042), intention to breastfeed (p= 0.002) and willingness to breastfeed (p= 0.005).

Comments and inquiries below are mostly minor: Please see the attached document:

1. Could you include how the participants were recruited? Did they receive hard copy, or they answered orally? Explain the method.

Participants were recruited when they were attending the maternity ward after delivery (as explained in data collection section).

“Medical students were responsible for data collection, where they asked mothers questions and mothers responded. In compliance with infection control procedures, and for avoiding paper-based questionnaires, medical students entered the answers electronically immediately, using their mobile phones or tablets”. 

2. Was there a complete lockdown in the delivery section? If yes, mention the period.

Lockdown occurred on and off according to covid-19 cases detected among patients or health care providers. 

3. Could you state the inclusion or exclusion criteria for selecting the participants?

Women who were still under the effect of anesthesia, or who seemed very tired at the time of data collection were excluded. 

4. Was there an informed consent or not?

Verbal consent was obtained from women as mentioned in the ethical considerations section.

5. Did you use any sampling method? Like random or systemic sampling technique for selecting the participants.

Convenience sample was used. All women available at the time of data collection were invited to participate.

6. It looks like interview. Is that right? If yes, did all the patients answered verbally? How long was it take? If mixed, written or interview, what was the percentage of each. Did you record the interview?

It was a structures interviewer-administered questionnaire. 

Medical students were responsible for data collection, where they asked mothers questions and mothers responded. In compliance with infection control procedures, and for avoiding paper-based questionnaires, medical students entered the answers electronically immediately, using their mobile phones or tablets. 

Interview was not recorded.

7. In the data collection procedure section: line 9: Medical students entered the answers electronically. Do you mean for analyzing the data? Clarify.

They asked the women questions, and entered the answers into the electronic questionnaire.

8. Did you take the permissions from the patients before collecting these data? Explain

“The research team attended the maternity ward whenever they were allowed to do so. They approached all mothers available in the ward at the time of their visit. They introduced themselves, and offered women the opportunity to participate in the research study. Women who were willing to participate were given further details and a verbal consent was obtained. “

9. In the analysis section: In addition, the mean and standard deviation for individual items of IIFAS, and total score for the attitude toward breastfeeding for each respondent were calculated. How? Which test did you use?

There was not test for calculating the mean. 

In table 3, the mean for individual item was presented.

In table 4, the total attitude score for each respondent was calculated by adding up the scores for the 17 items for each respondent.

10. In table 4: undetermined: Why, what was the reason? explain that in the footnote under the table.

A footnote was added beneath the table indicating what undetermined means.

11. Why are mothers more likely to use formula than breastfeed?

I have added: “ This may be due to the belief that formula milk is an advanced combination of nutrients and that it is as good as breastmilk.”

12. In the end of page 14: healthcare provider: Did they use ads for breastfeeding? What kind of encourage did they provide?

They remind women and bring their attention to the importance of breastfeeding, and encourage them to initiate breastfeeding and continue. 

Reviewer #5: Comment” The characteristics of the study participants should be mentioned as inclusion criteria and exclusion. The authors responded “There were no inclusion or exclusion criteria” . It cannot be, there must be inclusion and exclusion criteria for every 

“A convenience sample was selected from the target population for the current study, which was all postnatal women attending the maternity ward after delivery, when the research team was available in the hospital, during the study period. Women who were still under the effect of anesthesia, or who seemed very tired at the time of data collection were excluded. study and yours of no exception “.

Other comments by reviewers were endorsed adequately by the authors.

Further Comments:

Under Introduction: the statement “Yet, About 44% of infants 0–6 months old are exclusively breastfed over the period of 2015-2020 [4]. Where? I cannot access the link for reference 4.

I did copy and paste the link and it opened with me. In the Global Health Observatory, chose the early initiation of breastfeeding indicator and it displays the statistics. 

Under Introduction: the statement “The decision about infant feeding seems to be made before childbirth” the authors quoted reference [21], however this conclusion was not mentioned in that article. Please revise.

Naja et al. in their paper studied women’s “intention to breastfeed” using six questions. Their results indicated: 

“For the intention to breastfeed, 43.4% of women had very strong intentions to breastfeed, 25.4% had strong intentions, 18.4% fair intentions and 12.7% weak intentions (Fig. 4d).”

“Both positive attitude towards breastfeeding and strong intention to breastfeed were associated with EBF at four months, breastfeeding at four months, and breastfeeding at six months (Table 6).”

The objective of the study should be stated clearly at the end of the introduction.

This i“Hence, in this study, we sought to assess factors that determine women’s attitude towards breastfeeding in Jordan.”

What was the validity and reliability of IIFAS questionnaire? 

The IIFAS was prevalidated, and pilot test was performed and reliability was calculated (reference 28).

Item 17 of IIFAS questionnaire is about drinking alcohol for breastfeeding women. I find it difficult to ask Muslim Jordanian women this question. Please address my concern.

We did actually hesitate to keep this item about drinking alcohol, but then preferred to keep it. Since it can provide an insight into women’s perception about the influence of what they eat or drink on their infants. 

The authors stated that “ The researcher used a validated Arabic version of the IIFAS was used after permission from the author” and they quoted reference 29 for this statement. Indeed, the authors under reference 29 used a semi structured questionnaire survey and they did not use IIFAS questionnaire, please revise.

You are totally right. I got permission from Charafedding from Lebanon, who did the validation for the Arabic version.

Under data collection procedure: the authors sated that “response rate was 95% with 301 mothers completing the questionnaires” What was the dominator for 301 mothers?

The total number of women approached was 316, out of which, 15 did not want to participate. Leading to a 95% response rate.

The authors stated that “monthly family income of 450 Jordan Dinar or less” please convert this to US$ as readers are unfamiliar with Jordan Dinar.

I added a foot note beneath table 1 (1 JD equals US$ 1.4)

Reviewer #6: General comment:

Based on other studies on the determinants of breastfeeding, some of which are included in the manuscript bibliography, my question to the authors is: why was “knowledge of breastfeeding” not included in the study question? The Iowa Infant Feeding Attitude Survey (IIFAS) questionnaire used for data collection has items that can be better classified as “knowledge” rather than as “attitude”. The IIFAS tool developers could be asked as to whether they intentionally classified all survey items as assessment of “attitude” when in fact they appear to be testing “knowledge”.

While some questions may seem like knowledge oriented, I think they do measure perceptions and attitudes more accurately. 

Parturient mothers may be less intimidated to answer questions on their “attitude” and may be more apt to accept participation in the study, as opposed to thinking that they would be tested and evaluated with a survey of their correct or incorrect “knowledge,” whereas attitudes are neither right or wrong. However, the research needs to determine if the items are truly measuring “attitude” or “knowledge.” The manuscript would be clearer to readers if the authors could discuss the issue of “knowledge” versus “attitude” in relation to intent and practice of breastfeeding, and why they have excluded the assessment of knowledge. There may be studies available in the literature that address that issue.

This study did not aim to assess knowledge from the beginning. Our objective was to assess attitude to breastfeeding, and the IIFAS was one of the best scales used. We thought that knowledge about benefits of breastfeeding may be good, but why is the practice of breastfeeding deteriorating? That’s why we thought about examining attitudes.

A related issue is the possible presence of myths and cultural beliefs related to breastfeeding practice, and how those affect breastfeeding attitudes. These do not seem to have been addressed in the manuscript, though these myths and beliefs are highly prevalent in many societies. The authors should discuss the possible presence or not of breastfeeding myths and cultural beliefs in their study population of Jordan.

I totally agree with you about the presence of some beliefs about breastfeeding in Jordan, which may be a good idea for a new research paper later on. This may be addressed as a limitation in this study and can be recommendation for future research also. 

Added to limitations: “And finally, cultural beliefs about breastfeeding were not addressed as factors that may influence women’s attitude toward breastfeeding in Jordan, which may be the focus of future research”. 

Specific comments:

Abstract.

Second sentence has a misspelling of “determine.”

It has been corrected. 

Methodology

Medical students were used to interview mothers for this study. What percentage of these were female? There are two intersecting issues. One is that same-sex (female) interviewers may be more likely to obtain truthful answers from female interviewees. 

Medical students who worked on data collection were all females. We have realized this issue and chose females.

The second is that the interviewees may be also intimidated by perceived better-educated and perceived higher-status medical students, and may provide answers that mothers perceive as the “correct answers” rather than their true attitudes or knowledge.

This is an inherent bias in a survey (social desirability bias), because of human nature. But we tried to reduce this by using neutrally worded sentences. In addition, anonymous questionnaires should have reduced this. 

Results

Table 3 should indicate that the items were taken from the IIFAS questionnaire with reference cited.

Title of the table was modified:

Table 3. Item scores of mothers’ postnatal attitudes to breastfeeding using the IIFAS (N=301)

I think there is no need to reference the IIFAS, since all details were provided in the methods section. 

At the bottom of Table 3, the label should read “Overall mean score on attitudes toward infant feeding = 3.95 + 1.04”.

Noted. This has been corrected. 

Readers may likely wonder why an attitude score of 3.95 + 1.04 is considered by the authors as a “neutral” attitude. This designation should be reconsidered.

The cutoff points for attitude to breastfeeding (positive to breastfeeding, neutral, and negative to breastfeeding). This has been frequently used in the literature using the IIFAS, and it has been detailed in the methods section. 

Recommendations

Recommendations are primarily to do more studies on “attitudes.” My question again is related to the issue of breastfeeding “knowledge.” Are more studies needed on breastfeeding knowledge as well as on cultural beliefs and myths related to breastfeeding?

A recommendation about future research on breastfeeding knowledge was added.

Reviewer #7: The paper is generally well written and structured. Overall, this is a clear, concise, and well-written manuscript. The introduction is relevant. Sufficient information about the previous study findings is presented for readers.

Thank you. No other comments?

---

## [Decision Letter · Decision Letter 2]

3 Apr 2023

PONE-D-22-35675R2Determinants of Breastfeeding Attitudes of Mothers in Jordan: A Cross-sectional StudyPLOS ONE

Dear Dr. Alkhaldi,

Thank you for submitting your manuscript to PLOS ONE. After careful consideration, we feel that it has merit but does not fully meet PLOS ONE’s publication criteria as it currently stands. Therefore, we invite you to submit a revised version of the manuscript that addresses the points raised during the review process.

We look forward to receiving your revised manuscript.

Kind regards,

Ghada Abdrabo Abdellatif Elshaarawy, M.D

Academic Editor

PLOS ONE

Journal Requirements:

Additional Editor Comments:

Reviewer #4 comment: In table 4, What was the reason for undetermined? explain that in the footnote under the table.

Make sure you thoroughly address the reviewer’s #6 comments and concerns.

Reviewers' comments:

Reviewer's Responses to Questions

**Comments to the Author**

1. If the authors have adequately addressed your comments raised in a previous round of review and you feel that this manuscript is now acceptable for publication, you may indicate that here to bypass the “Comments to the Author” section, enter your conflict of interest statement in the “Confidential to Editor” section, and submit your "Accept" recommendation.

Reviewer #4: All comments have been addressed

Reviewer #5: All comments have been addressed

Reviewer #6: (No Response)

Reviewer #7: All comments have been addressed

2. Is the manuscript technically sound, and do the data support the conclusions?

Reviewer #4: Yes

Reviewer #5: Yes

Reviewer #6: No

Reviewer #7: Yes

3. Has the statistical analysis been performed appropriately and rigorously? 

Reviewer #4: Yes

Reviewer #5: Yes

Reviewer #6: No

Reviewer #7: Yes

4. Have the authors made all data underlying the findings in their manuscript fully available?

Reviewer #4: No

Reviewer #5: Yes

Reviewer #6: Yes

Reviewer #7: (No Response)

5. Is the manuscript presented in an intelligible fashion and written in standard English?

Reviewer #4: Yes

Reviewer #5: Yes

Reviewer #6: Yes

Reviewer #7: Yes

6. Review Comments to the Author

Reviewer #4: (No Response)

Reviewer #5: Thank you very much for addressing my comments . The required corrections have been made and the manuscript has been improved scientifically. The authors responded to my comment and provided valuable information which are important for the readers. l

Reviewer #6: Page 2. This paper reports on a study conducted on recently-delivered mothers in a Jordan tertiary-level hospital among whom 75% had university or higher education. The authors should provide data on the average educational level of Jordanian women to provide a better understanding of the generalizability of this study. If the authors intended to help explain the low rate of exclusive breastfeeding in Jordan in order to make policy recommendations, a representative selection of study mothers would have been a better choice. The generalizability of findings needs to be prominently discussed at the end of the paper.

In the introduction, the authors cite literature that higher education is associated with a good or positive attitude toward breastfeeding. The authors should try to explain why the study had measured only an average score of “neutral attitude” toward breastfeeding even though such a high proportion of their study mothers had a high level of education. They should explain in the discussion section that this study has dissimilar results to findings of other studies on the issue of higher education mothers, as they found no association of education with the breastfeeding attitude score.

Authors should also note that although highest income mothers had a significantly better attitude score as compared to the lowest income mothers, medium low-level income mothers were also significantly different from lowest income. Discuss the implications of this finding.

The average score of 65 + 7 out of 85 on the IIFAS survey questionnaire was very close to the upper limit of the neutral range 49-69, and could be stated that the tendency was towards a positive attitude.

The authors cite a study by Khasawneb (2020) that found 72% of mothers with a positive attitude toward breastfeeding. The authors should discuss why their findings are so different with only 24% with a positive attitude.

Page 3. After discussing the importance of exclusive breastfeeding, the authors state that the ‘breastfeeding’ rate in Jordan is 25% and compares this to similar rates reported in other countries. The authors should specify if they are referring to exclusive breastfeeding rates as 25%, or to “any breastfeeding” in children 0-5 months of age.

Page 4. In the Introduction, the authors cite two studies conducted in Jordan (2006 and 2018) which reported that positive maternal attitudes toward breastfeeding were associated with exclusive breastfeeding. The authors should provide more details on the socio-demographic-economic characteristics of those two study populations, and also what instruments had been used to measure positive maternal attitude. This would allow the authors to provide a more complete justification of the need for their current study, for their choice of study population, and for their choice of the IIFAS as the attitude measurement instrument.

In order for readers to better understand the current research, the paper would benefit from a discussion of the concept of “attitudes”: what exactly is an attitude, types of attitudes, and whether attitudes are predictive of behavior. The same suggestion is made for the concept of “ intentions”, since 95% of the study population had the intention to breastfeed (see Table 2). This discussion should include an explanation of the differences between “attitudes” and “intentions,” as well as the differences between “intentions” and “behaviors”.

The data on Table 2 should be included in the sections on results, discussion, conclusions, and recommendations. Based on data reported on Table 2, it appears that the hospital is not carrying out a strong role in promoting positive breastfeeding behaviors. Only 43% of study mothers had breastfed in the first day after birth. Also, only 40% of mothers had received breastfeeding counseling (20% prenatal and 20% postnatal) and 60% had received no counseling. This was in spite of the authors’ statement that the study location hospital was an “accredited Baby Friendly Hospital” since 2018.

In regard to the 43% of breastfeeding on Day 1, the correct WHO definition of early initiation of breastfeeding (EIBF) is “provision of mothers' breast milk to infants within the first hour of birth.” EIBF is the breastfeeding indicator that is most predictive of successful breastfeeding and reduced neonatal and infant morbidity and mortality. If the study questionnaire had included the question on initiation of breastfeeding in the first hour of life, this should be reported in Table 2. Breastfeeding on the first day is not an accepted breastfeeding indicator.

Table 4 shows the average score of “attitude toward breastfeeding” (based on the IIFAS) for each category of all the study variables. It is notable that the mean attitude is nearly the same for all categories of all variables. Even for variables such as Working Status of the mother which is mentioned earlier in the paper as a significant factor associated with breastfeeding intentions and practices, the average IIFAS score is exactly the same for both employed and unemployed mothers. The authors should explain why the attitude scores are so very similar for nearly all categories of all variables. One could wonder if the IIFAS is able to detect a difference in attitudes in this study population. If not, the IIFAS could be not really a valid test for this population. On the other hand, if the IIFAS is accurately measuring attitudes toward breastfeeding, then one could conclude that all Jordanian mothers in this population have very similar attitudes toward breastfeeding and in which case, therefore, attitudes are unlikely to be associated with breastfeeding practice. Contrary to findings in this study, in many country studies that could be cited, lower income women are more likely to breastfeed and to exclusively breastfeed for a longer period. That contradiction should be explained.

Statistics

Table 5 with logistic regression results should refer to factors that are “associated” with the attitude score. A cross-sectional observational study cannot infer causality as suggested by the term “predictors.” This issue could extend to the title of the study that uses the term “Determinants.” Table 5 should include a list of control variables beneath the table, if a step-wise regression was used. Alternatively, the table should include OR and CI results of the entire regression model with all variables included that had had <.10 p value in the bivariate analyses. And next to that, the best-fitting model. Each with r and r2 values reported.

Regarding the data presented on Table 5, there do not seem to be clear results on the variables of “monthly family income” and “willingness to breastfeed exclusively.”

On Table 5, the variable on “monthly family income” has 4 levels from low (1) to high (4). Levels 2 and 4 are significant. Level 3 is not significant. These findings cannot be interpreted as only the highest income mothers having significantly better attitudes toward breastfeeding. Rather, the level 2 of income is also significantly different from level 1. (OR 4.72, CI 1.12 – 26.86). The p-level is less important than the CI in regressions.

On Table 5, the variable “willingness to breastfeed exclusively” has confusing confidence intervals. It seems the confidence intervals are inverted for the categories “yes” and “undetermined” and should be reviewed and corrected if necessary.

Conclusions

Conclusions are not supported by data. The discussion refers to the positive associations of “pregnancy complications,” “delivery complications,” and “intention to breastfeed” with the outcome of attitude score. However, the former variables (pregnancy and delivery complications) have non-significant associations in the logistic regression. The latter variable (intention to breastfeed) is not reported in the logistic regression on Table 5.

Conclusions should be modest and not attempt to go beyond the findings of the study. The study does not show that the attitudes measured are predictive of breastfeeding practice. You will not be able to make conclusions and recommendations about how to improve breastfeeding practice as your data does not provide information on that issue.

Recommendations

Recommendations should suggest conducting a study that looks at whether the IIFAS is associated with breastfeeding practices in Jordan, controlling for other factors such as hospital breastfeeding routines (early initiation of breastfeeding, rooming-in, etc), newborn characteristics (prematurity, birth weight, birth by cesarean section, etc) and maternal characteristics (educational level, work status, income, etc). This would be a way to test the concept validity of the IIFAS for the Jordanian population of parturient mothers as a predictor of breastfeeding practice. This study would ideally be conducted on a representative population sample with a prospective design so that results can contribute to policy recommendations for the Jordanian population.

Reviewer #7: The paper is generally well written and structured. Overall, this is a clear, concise, and well-written manuscript.

7. PLOS authors have the option to publish the peer review history of their article (what does this mean?). If published, this will include your full peer review and any attached files.

Reviewer #4: No

Reviewer #5: No

Reviewer #6: **Yes: **Laura Altobelli

Reviewer #7: **Yes: **MAHMOUD AL-MASAEED

---

## [Author Response · Author response to Decision Letter 2]

7 Apr 2023

PONE-D-22-35675R2

Determinants of Breastfeeding Attitudes of Mothers in Jordan: A Cross-sectional Study

PLOS ONE

Journal Requirements:

Additional Editor Comments:

Reviewer #4 comment: In table 4, What was the reason for undetermined? explain that in the footnote under the table.

Response: I have explained the meaning of undetermined in a footnote under the table.

Make sure you thoroughly address the reviewer’s #6 comments and concerns.

Reviewers' comments:

Reviewer's Responses to Questions

Comments to the Author

1. If the authors have adequately addressed your comments raised in a previous round of review and you feel that this manuscript is now acceptable for publication, you may indicate that here to bypass the “Comments to the Author” section, enter your conflict of interest statement in the “Confidential to Editor” section, and submit your "Accept" recommendation.

Reviewer #4: All comments have been addressed

Reviewer #5: All comments have been addressed

Reviewer #6: (No Response)

Reviewer #7: All comments have been addressed

6. Review Comments to the Author

Reviewer #4: (No Response)

Reviewer #5: Thank you very much for addressing my comments . The required corrections have been made and the manuscript has been improved scientifically. The authors responded to my comment and provided valuable information which are important for the readers. 

Reviewer #6:

 Page 2. This paper reports on a study conducted on recently-delivered mothers in a Jordan tertiary-level hospital among whom 75% had university or higher education. The authors should provide data on the average educational level of Jordanian women to provide a better understanding of the generalizability of this study. If the authors intended to help explain the low rate of exclusive breastfeeding in Jordan in order to make policy recommendations, a representative selection of study mothers would have been a better choice. The generalizability of findings needs to be prominently discussed at the end of the paper.

RESPONSE: The sample was selected from mothers attending this referral teaching hospital in the capital Amman. Many of women who receive service in this hospital are university employees and their families. This explains the high level of education among the participants. These women may not be representative of all women in Jordan. (This has been added to limitations).

In the introduction, the authors cite literature that higher education is associated with a good or positive attitude toward breastfeeding. The authors should try to explain why the study had measured only an average score of “neutral attitude” toward breastfeeding even though such a high proportion of their study mothers had a high level of education. They should explain in the discussion section that this study has dissimilar results to findings of other studies on the issue of higher education mothers, as they found no association of education with the breastfeeding attitude score.

RESPONSE: On the other hand, mothers with higher education did not show more positive attitude to breastfeeding. This was inconsistent with results of other studies in the literature, where higher education was significantly associated with more positive attitude to breastfeeding. This may be explained by the fact that mothers with higher education are working or planning to work. For these mothers, bottle-feeding may be considered a more convenient option for infant feeding, since it is less time consuming and can be provided by any care giver. These mothers may also value freedom, and prefer to choose infant feeding method that will not restrict their daily activities. 

Authors should also note that although highest income mothers had a significantly better attitude score as compared to the lowest income mothers, medium low-level income mothers were also significantly different from lowest income. Discuss the implications of this finding.

RESPONSE: Mothers with higher income level seem to care more about the quality of food they provide for their families, including choosing breastfeeding for their infants. Mothers with lowest income level demonstrated more positive attitude to formula milk feeding. These women cannot afford the price of formula milk, and have no choice but to breastfeed. This may explain why they favor formula milk more, and may perceive it as an ideal nutritious food for their infants. 

The average score of 65 + 7 out of 85 on the IIFAS survey questionnaire was very close to the upper limit of the neutral range 49-69, and could be stated that the tendency was towards a positive attitude.

RESPONSE: I agree with the reviewer. I restated the description as being close to the upper limit of the neutral range, with tendency toward a positive attitude. I have changed that in the results and in the abstract. 

The authors cite a study by Khasawneb (2020) that found 72% of mothers with a positive attitude toward breastfeeding. The authors should discuss why their findings are so different with only 24% with a positive attitude.

Response: Khasawneh et al (2020) have overestimated the proportion of mothers with positive attitude to breastfeeding. They offered the choice of only agree or disagree for the IIFAS scale, and agreement with more than half of the items was considered as positive attitude to breastfeeding. 

Page 3. After discussing the importance of exclusive breastfeeding, the authors state that the ‘breastfeeding’ rate in Jordan is 25% and compares this to similar rates reported in other countries. The authors should specify if they are referring to exclusive breastfeeding rates as 25%, or to “any breastfeeding” in children 0-5 months of age.

Response: It was not exclusive breastfeeding, it was any breastfeeding. It has been clarified in page 3.

Page 4. In the Introduction, the authors cite two studies conducted in Jordan (2006 and 2018) which reported that positive maternal attitudes toward breastfeeding were associated with exclusive breastfeeding. The authors should provide more details on the socio-demographic-economic characteristics of those two study populations, and also what instruments had been used to measure positive maternal attitude. This would allow the authors to provide a more complete justification of the need for their current study, for their choice of study population, and for their choice of the IIFAS as the attitude measurement instrument.

Response: The study population in Khasawneh et. al. [24] was a randomly selected 344 women from the North of Jordan villages. Using questions adapted from the IIFAS. While Khasawneh et. al. [27] interviewed healthy women attending antenatal clinics in three hospitals in the North of Jordan too, using the IIFAS. 

In order for readers to better understand the current research, the paper would benefit from a discussion of the concept of “attitudes”: what exactly is an attitude, types of attitudes, and whether attitudes are predictive of behavior. The same suggestion is made for the concept of “ intentions”, since 95% of the study population had the intention to breastfeed (see Table 2). This discussion should include an explanation of the differences between “attitudes” and “intentions,” as well as the differences between “intentions” and “behaviors”.

Response: 

Attitude here is defined as a mental position with regard to a fact or state, and the feeling or emotion toward the fact or state. Attitude is traditionally structured along three dimensions: cognitive, affective, and behavioral. Intention is a prior conscious decision to perform a behavior. Intention can reflect maternal commitment to infant’s health.

The breastfeeding behavior is directly determined by the intention to breastfeed. Intention is formed through a combination of attitudes, subjective norm, and perceived behavioural control.

The data on Table 2 should be included in the sections on results, discussion, conclusions, and recommendations. Based on data reported on Table 2, it appears that the hospital is not carrying out a strong role in promoting positive breastfeeding behaviors. Only 43% of study mothers had breastfed in the first day after birth. Also, only 40% of mothers had received breastfeeding counseling (20% prenatal and 20% postnatal) and 60% had received no counseling. This was in spite of the authors’ statement that the study location hospital was an “accredited Baby Friendly Hospital” since 2018.

Response: 

Discussion: Despite the fact that this hospital is a baby friendly hospital, only 43% of mothers initiated breastfeeding in the first day after childbirth, and 60% of mothers did not receive any counselling of advice related to breastfeeding. All this, given that 97.5% of the mothers had the intention to breastfeed. This may be explained by the hospital being a tertiary referral one, receiving a large volume of complicated cases requiring cesarean section, and inducing delay in initiation of breastfeeding. On the other hand, the role of this hospital in promoting positive breastfeeding behavior is suboptimal. 

Mothers who delivered in this hospital, did not necessarily receive antenatal care in this hospital. 

In recommendations: In addition, hospitals that are designated as baby friendly, must audit their procedures related to encouraging breastfeeding practice, and make sure that related policies are properly implemented.

In conclusions: Healthcare professionals at all level of the health care system must play their vital role in promoting breastfeeding, starting with prenatal counselling, and ending with immediate postnatal counselling and supporting mothers to initiate breastfeeding their infants. All this in the attempt to increase to the rates of breastfeeding among women in Jordan.

In regard to the 43% of breastfeeding on Day 1, the correct WHO definition of early initiation of breastfeeding (EIBF) is “provision of mothers' breast milk to infants within the first hour of birth.” EIBF is the breastfeeding indicator that is most predictive of successful breastfeeding and reduced neonatal and infant morbidity and mortality. If the study questionnaire had included the question on initiation of breastfeeding in the first hour of life, this should be reported in Table 2. Breastfeeding on the first day is not an accepted breastfeeding indicator.

Response: Data on initiation in the first hour after delivery is not available. It was not the focus of this study. 

Table 4 shows the average score of “attitude toward breastfeeding” (based on the IIFAS) for each category of all the study variables. It is notable that the mean attitude is nearly the same for all categories of all variables. Even for variables such as Working Status of the mother which is mentioned earlier in the paper as a significant factor associated with breastfeeding intentions and practices, the average IIFAS score is exactly the same for both employed and unemployed mothers. The authors should explain why the attitude scores are so very similar for nearly all categories of all variables. One could wonder if the IIFAS is able to detect a difference in attitudes in this study population. If not, the IIFAS could be not really a valid test for this population. On the other hand, if the IIFAS is accurately measuring attitudes toward breastfeeding, then one could conclude that all Jordanian mothers in this population have very similar attitudes toward breastfeeding and in which case, therefore, attitudes are unlikely to be associated with breastfeeding practice. Contrary to findings in this study, in many country studies that could be cited, lower income women are more likely to breastfeed and to exclusively breastfeed for a longer period. That contradiction should be explained.

Response: Average attitude scores did not differ across different categories, because there were nor real differences in their attitudes.

Still, we found attitude to be not associated with actual practice of initiating breastfeeding (and we have discussed this). 

Our results did indicate that poor women had less positive attitude to breastfeeding. But it was not the focus of this study to analyze the relationship between income level and practice of breastfeeding. 

Statistics

Table 5 with logistic regression results should refer to factors that are “associated” with the attitude score. A cross-sectional observational study cannot infer causality as suggested by the term “predictors.” This issue could extend to the title of the study that uses the term “Determinants.” 

This has been changed.

Table 5 should include a list of control variables beneath the table, if a step-wise regression was used. Alternatively, the table should include OR and CI results of the entire regression model with all variables included that had had <.10 p value in the bivariate analyses. And next to that, the best-fitting model. Each with r and r2 values reported.

Response: Results have been revisited and corrected as requested.

Regarding the data presented on Table 5, there do not seem to be clear results on the variables of “monthly family income” and “willingness to breastfeed exclusively.”

Response: Results have been revisited, variables added as suggested and finalized.

On Table 5, the variable on “monthly family income” has 4 levels from low (1) to high (4). Levels 2 and 4 are significant. Level 3 is not significant. These findings cannot be interpreted as only the highest income mothers having significantly better attitudes toward breastfeeding. Rather, the level 2 of income is also significantly different from level 1. (OR 4.72, CI 1.12 – 26.86). The p-level is less important than the CI in regressions.

Response: interpretation was corrected.

On Table 5, the variable “willingness to breastfeed exclusively” has confusing confidence intervals. It seems the confidence intervals are inverted for the categories “yes” and “undetermined” and should be reviewed and corrected if necessary.

Response: 95%CI have been reviewed and corrected. 

Conclusions

Conclusions are not supported by data. The discussion refers to the positive associations of “pregnancy complications,” “delivery complications,” and “intention to breastfeed” with the outcome of attitude score. However, the former variables (pregnancy and delivery complications) have non-significant associations in the logistic regression. The latter variable (intention to breastfeed) is not reported in the logistic regression on Table 5.

Response: Discussion and Conclusions have been modified in order to match the results of this study.

Conclusions should be modest and not attempt to go beyond the findings of the study. The study does not show that the attitudes measured are predictive of breastfeeding practice. You will not be able to make conclusions and recommendations about how to improve breastfeeding practice as your data does not provide information on that issue.

Response: Recommendations have been aligned with results of this study.

Recommendations

Recommendations should suggest conducting a study that looks at whether the IIFAS is associated with breastfeeding practices in Jordan, controlling for other factors such as hospital breastfeeding routines (early initiation of breastfeeding, rooming-in, etc), newborn characteristics (prematurity, birth weight, birth by cesarean section, etc) and maternal characteristics (educational level, work status, income, etc). This would be a way to test the concept validity of the IIFAS for the Jordanian population of parturient mothers as a predictor of breastfeeding practice. This study would ideally be conducted on a representative population sample with a prospective design so that results can contribute to policy recommendations for the Jordanian population.

Response: 

We suggest that future studies look at whether the IIFAS is associated with breastfeeding practices in Jordan, controlling for other factors such as hospital breastfeeding routines (early initiation of breastfeeding, rooming-in, etc), newborn characteristics (prematurity, birth weight, birth by cesarean section, etc) and maternal characteristics (educational level, work status, income, etc).

We also recommend that future studies be conducted on a representative population sample with a prospective design so that results can contribute to policy recommendations for the Jordanian population.

---

## [Editor Report · Decision Letter 3]

14 Apr 2023

PONE-D-22-35675R3Determinants of Breastfeeding Attitudes of Mothers in Jordan: A Cross-sectional StudyPLOS ONE

Dear Dr. Alkhaldi,

Thank you for submitting your manuscript to PLOS ONE. After careful consideration, we feel that it has merit but does not fully meet PLOS ONE’s publication criteria as it currently stands. Therefore, we invite you to submit a revised version of the manuscript that addresses the points raised during the review process.

We look forward to receiving your revised manuscript.

Kind regards,

Ghada Abdrabo Abdellatif Elshaarawy, M.D

Academic Editor

PLOS ONE

Journal Requirements:

Additional Editor Comments:

The correct WHO definition of early initiation of breastfeeding (EIBF) is “provision of mothers' breast milk to infants within the first hour of birth.” EIBF is the breastfeeding indicator that is most predictive of successful breastfeeding and reduced neonatal and infant morbidity and mortality.

Since the data on initiation of breastfeeding in the first hour after delivery is not available, so you should add this to the limitations of the study.

---

## [Author Response · Author response to Decision Letter 3]

14 Apr 2023

PONE-D-22-35675R3

Determinants of Breastfeeding Attitudes of Mothers in Jordan: A Cross-sectional Study

PLOS ONE

Additional Editor Comments:

The correct WHO definition of early initiation of breastfeeding (EIBF) is “provision of mothers' breast milk to infants within the first hour of birth.” EIBF is the breastfeeding indicator that is most predictive of successful breastfeeding and reduced neonatal and infant morbidity and mortality.

Since the data on initiation of breastfeeding in the first hour after delivery is not available, so you should add this to the limitations of the study.

Response: I have added to the limitations about the Lack of data regarding initiation of breastfeeding in the first hour after delivery.

---

## [Editor Report · Decision Letter 4]

24 Apr 2023

Determinants of Breastfeeding Attitudes of Mothers in Jordan: A Cross-sectional Study

PONE-D-22-35675R4

Dear Dr. Alkhaldi,

We’re pleased to inform you that your manuscript has been judged scientifically suitable for publication and will be formally accepted for publication once it meets all outstanding technical requirements.

Kind regards,

Ghada Abdrabo Abdellatif Elshaarawy, M.D

Academic Editor

PLOS ONE
---

## [Editor Report · Acceptance letter]

28 Apr 2023

PONE-D-22-35675R4 

Determinants of Breastfeeding Attitudes of Mothers in Jordan:  A Cross-sectional Study 

Dear Dr. Alkhaldi:

I'm pleased to inform you that your manuscript has been deemed suitable for publication in PLOS ONE. Congratulations! Your manuscript is now with our production department. 

Kind regards, 

on behalf of

Dr. Ghada Abdrabo Abdellatif Elshaarawy 

Academic Editor

PLOS ONE